# Intraocular liver spheroids for non-invasive high-resolution in vivo monitoring of liver cell function

Francesca Lazzeri-Barcelo[1], Nuria Oliva-Vilarnau[2], Marion Baniol[3], Barbara Leibiger[1], Olaf Bergmann [3,4,5], Volker M. Lauschke [2,6,7], Ingo B. Leibiger[1], Noah Moruzzi [1,10] ✉ & Per-Olof Berggren[1,8,9,10]

Longitudinal monitoring of liver function in vivo is hindered by the lack of high-resolution non-invasive imaging techniques. Using the anterior chamber of the mouse eye as a transplantation site, we have established a platform for longitudinal in vivo imaging of liver spheroids at cellular resolution. Transplanted liver spheroids engraft on the iris, become vascularized and innervated, retain hepatocyte-specific and liver-like features and can be studied by in vivo confocal microscopy. Employing fluorescent probes administered intravenously or spheroids formed from reporter mice, we showcase the potential use of this platform for monitoring hepatocyte cell cycle activity, bile secretion and lipoprotein uptake. Moreover, we show that hepatic lipid accumulation during diet-induced hepatosteatosis is mirrored in intraocular in vivo grafts. Here, we show a new technology which provides a crucial and unique tool to study liver physiology and disease progression in pre-clinical and basic research.

The health burden of liver diseases in industrialized countries is steadily increasing. Among those, lifestyle-related liver disorders are predicted to overtake viral hepatitis and become the leading cause of chronic liver disease in the Western world[1]. The research field focusing on the pathophysiological mechanisms of liver diseases and regeneration has hugely advanced over the last decades. Three-dimensional culture models, such as liver organoids and spheroids, have replaced monolayer and sandwich culture techniques due to the higher retention of hepatocyte differentiation by cell–cell interactions and cell polarization[2,3]. Moreover, co-culture systems with non-parenchymal liver cells, such as Kupffer and hepatic stellate cells, have improved spheroid functions to more closely mimic in situ hepatocyte features and functions[4,5]. Despite the advantages of allowing high-throughput experiments, these in vitro setups are limited due to the lack of

anatomical features, time constraints, hepatocyte de-differentiation, and the absence of vascularization and innervation. Although a recent study achieved vascularization of human liver spheroids on a microfluidic vascular bed improving hepatic tissue modeling[6], a key limitation of current in vitro liver models is the lack of a physiological environment. Replacing innervation has so far not been attempted in vitro, although recent studies have uncovered its complex role in an array of different liver functions[7].

Liver spheroids generated in vitro have been successfully transplanted into recipient mice, however, these sites of engraftment, such as under the kidney capsule[8], intraperitoneally[9], and subcutaneously[8] are not accessible for optical imaging. The state-of-the-art technique to be able to monitor liver parameters at a cellular level is intravital imaging, which allows high-resolution

[1]The Rolf Luft Research Center for Diabetes and Endocrinology, Karolinska Institutet, Stockholm, Sweden. [2]Department of Physiology and Pharmacology, Karolinska Institutet, Stockholm, Sweden. [3]Department of Cell and Molecular Biology, Karolinska Institutet, Stockholm, Sweden. [4]Center for Regenerative Therapies Dresden, TU Dresden, Dresden, Germany. [5]Department of Pharmacology and Toxicology, University Medical Center Goettingen, Goettingen, Germany. [6]Dr. Margarete Fischer-Bosch Institute of Clinical Pharmacology, Stuttgart, Germany. [7]University of Tübingen, Tübingen, Germany. [8]Tecnológico de Monterrey, Monterrey, NL, Mexico. [9]Diabetes Research Institute, University of Miami. Miller School of Medicine, Miami, Fl, USA. [10]These authors contributed equally: Noah Moruzzi, Per-Olof Berggren. ✉e-mail: noah.moruzzi@ki.se

visualization of the in situ liver. However, this technique is often terminal and does not allow regular imaging sessions of the same animal. Alternatively, the cornea is transparent and acts as a natural body window, through which microtissues engrafted on the iris can be imaged in a non-invasive and longitudinal manner by confocal microscopy[10–12], possibly solving this bottleneck in liver research. Therefore, the anterior chamber of the eye (ACE), with its highly vascularized and innervated iris, can be an ideal site for the engraftment and imaging of transplanted liver spheroids.

In this work, we have established an in vivo imaging platform aimed to overcome the major hurdles in methodology that largely limit liver research, allowing non-invasive longitudinal imaging of differentiated liver cells in a physiological milieu. By monitoring cellular function in the ACE in real-time, longitudinally, and at single-cell resolution, we now demonstrate the application of this platform in different areas of liver research.

## Results

### Liver spheroids engraft in the anterior chamber of the eye

To study hepatocyte functions longitudinally in vivo, we generated mouse liver spheroids and transplanted them into the ACE of recipient mice (Fig. 1a, b and Supplementary Fig. 1). Via injection through the cornea, the spheroids were positioned on the iris, where they engrafted and were imaged in real-time to generate high-resolution images in living animals (Fig. 1c−f).

Pivotal to this imaging platform, is the graft's ability to receive and respond to systemic cues through vascularization and innervation of the transplanted liver spheroids. To study vascularization, we performed in vivo imaging of fluorescently labeled lectin post-transplantation (post-tx). One-month post-tx, the vascular growth plateaus, and no further changes are evident thereafter (Fig. 2a−e and Supplementary Fig. 2a). At this point, the spheroid vascular network was formed by a mesh of uniformly sized capillaries (diameter

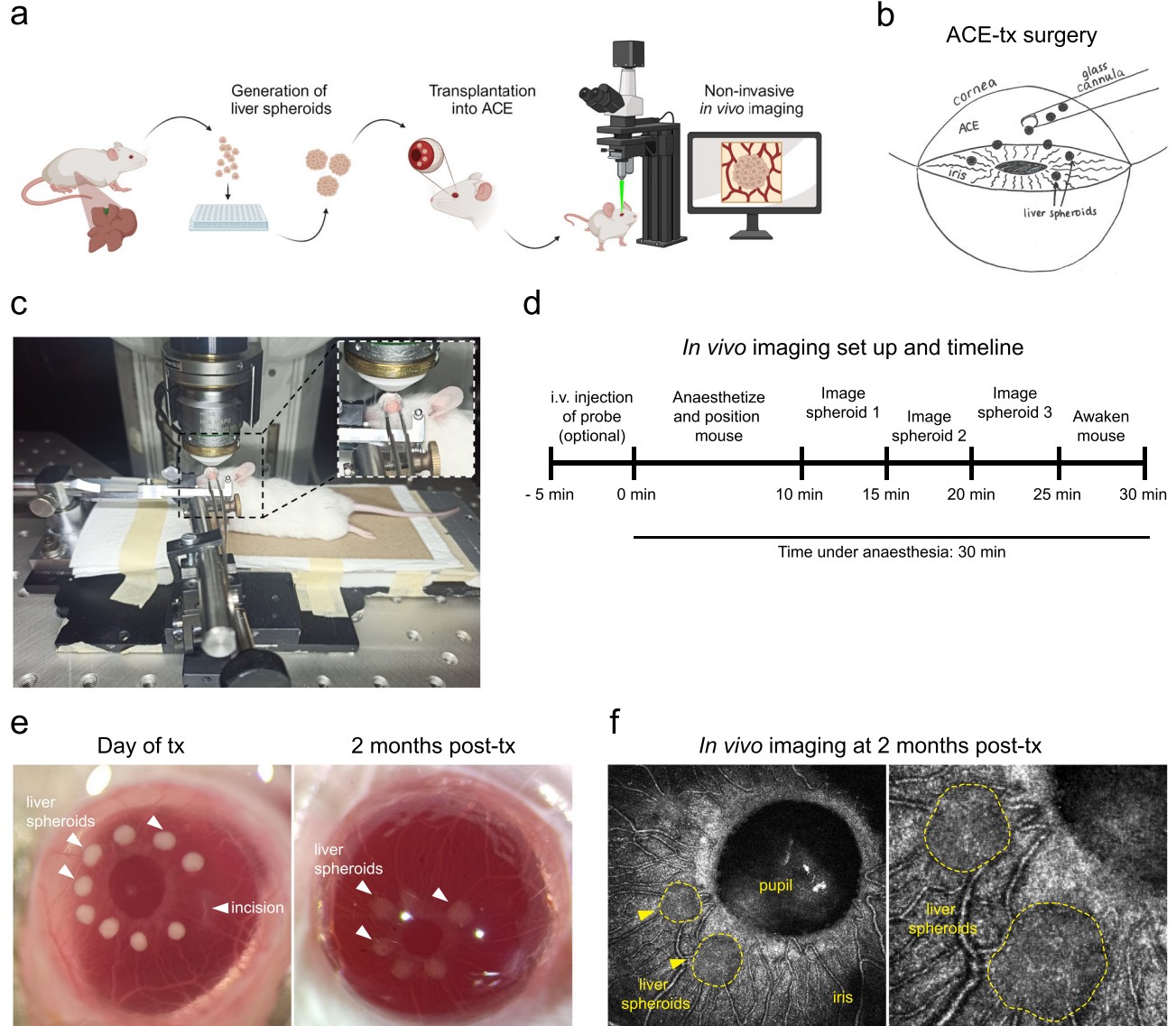

**Fig. 1 | The anterior chamber of the eye (ACE) as a transplantation site for longitudinal in vivo imaging of liver-like tissue. a** Experimental design of anterior chamber of the eye (ACE) in vivo imaging platform: liver spheroids were generated from primary liver cells (enriched for hepatocytes) and transplanted into the ACE of recipient mice, where they can be longitudinally imaged through the cornea by confocal microscopy. **b** Diagram of ACE transplantation (tx) surgery. **c** In vivo imaging setup and positioning of the mouse. **d** Experimental timeline of in vivo imaging session. **e** Stereoscopic images of liver spheroids (white arrowheads) transplanted into the ACE of albino recipient mice, on the day of tx and at 2 months post-tx. **f** Images of liver spheroids in vivo in anesthetized mice at 2 months post-tx, using label-free backscatter signal (white), Scale bar = 250 μm.

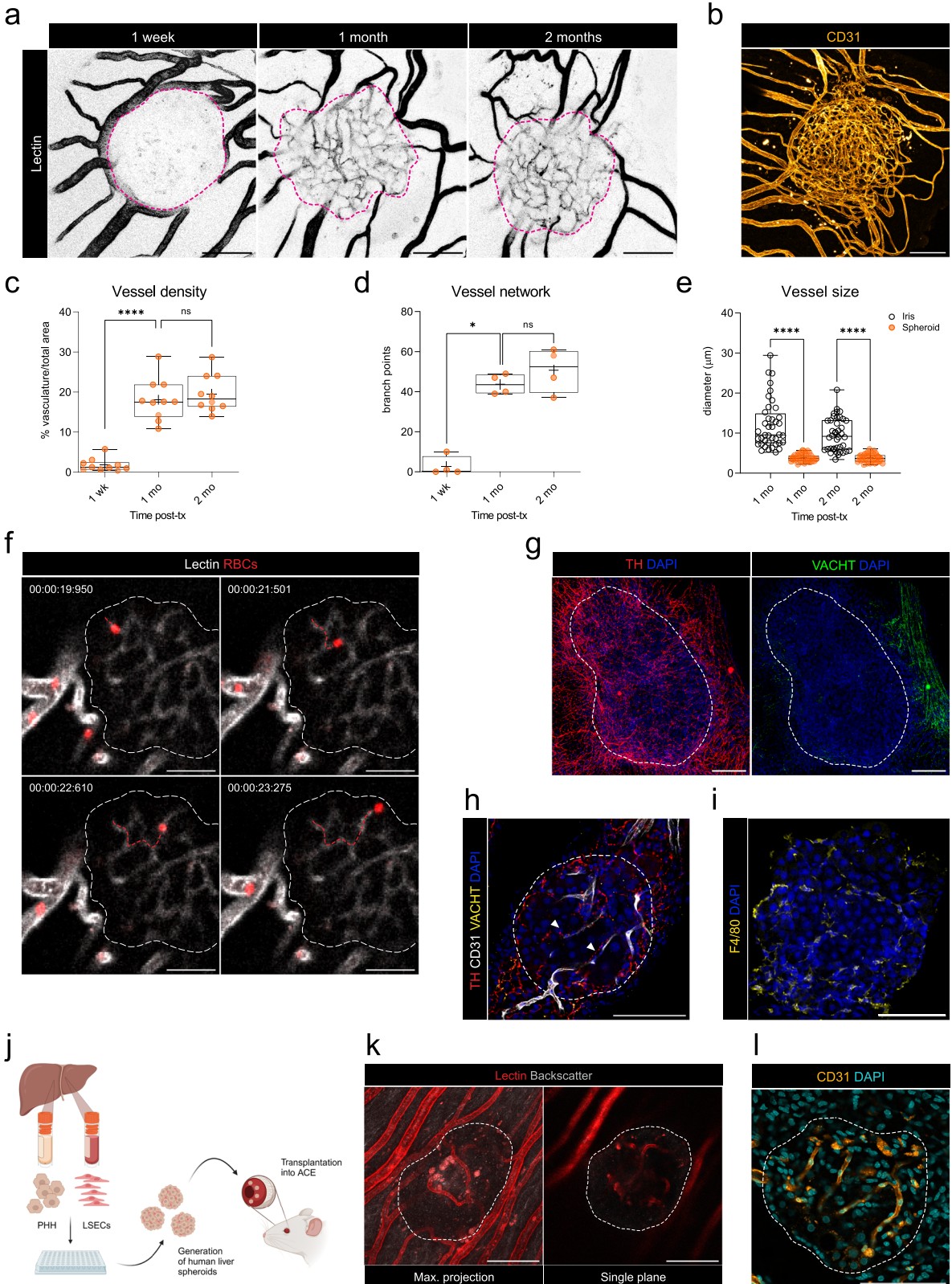

mean ± SD 3.8 ± 0.9 μm) approximately 3-fold smaller than those of the iris (Fig. 2e), indicating that the ingrowing vessel-type was dictated by the graft. This was confirmed by ex vivo immunofluorescence staining (Fig. 2b). By using labeled red blood cells (RBCs) during in vivo imaging, we found the vessels within the liver spheroids to be functional with unidirectional blood flow (Fig. 2f and Supplementary Movie 1). Innervation is another requirement for the liver spheroids to connect

to the recipient animal. Thus, we investigated innervation by ex vivo staining of sympathetic and parasympathetic markers (Fig. 2g and Supplementary Fig. 2b, c). Engrafted liver spheroids showed sympathetic innervation and nerve fibers can be seen running alongside vessels (Fig. 2h). Of note, the engrafted spheroids lacked parasympathetic innervation, despite both nerve types being present in the surrounding iris tissue (Fig. 2g, h). As for the vasculature, the

**Fig. 2 | Liver spheroids engraft and become vascularized and innervated in the ACE. a** Representative maximum projection of vessels in ACE-liver spheroid acquired by in vivo imaging by intravenous injection of DyLight-649-conjugated lectin (black) after binary conversion and thresholding at 1-week, 1 and 2-months post-tx. Spheroid area limit: pink dashed line. **b** Immunofluorescence vasculature network (CD31, orange) staining of whole-mount ACE-liver spheroid at 2-months post-tx. Max. projection image. **c** Vascular density of ACE-liver spheroids. Whiskers represent min/max values, with the mean shown as '+'. ****$p < 0.0001$ by one-way ANOVA test, $n = 9$ in 3 recipient mice. **d** Vessel network assessed by number of branch points within the spheroids. Whiskers represent min/max values, with the mean shown as '+'. *$p < 0.05$ by Mann–Whitney test, $n = 4$ in 3 recipient mice. **e** Average diameter of iris and intra-spheroid vessels at 1 and 2-months post-tx. Whiskers represent the min/max values, with the mean shown as '+'. ****$p < 0.0001$ by Kruskal–Wallis test, $n = 8$ measurements per spheroid, 5 spheroids in 3 recipient mice. **f** In vivo imaging of labeled red blood cells (red) through intra-spheroid vessels (lectin, white) and their trajectory (red-dashed line). Spheroid mass is delimited by a white-dashed line, scale bar = 50 μm.

Chronological images taken from Supplementary Movie 1. **g** Immunofluorescence staining of whole-mount ACE-liver spheroids and surrounding iris tissue. Sympathetic (TH, red) and parasympathetic (VACHT, green) nerves at 2-months post-tx. Max. projection images. **h** Immunofluorescence staining within the ACE-liver spheroid mass. Sympathetic (TH, red) and parasympathetic nerves (VACHT, yellow); vessels (CD31, white). Arrowheads indicate nerves alongside vessels and spheroid mass is delimited by white-dashed line. Single plane image.
**i** Immunofluorescence staining of macrophages (F4/80, yellow) within the ACE-liver spheroid. Single plane image. **j** Experimental outline of human liver spheroids generated from primary human hepatocytes (PHH) and primary human liver sinusoidal endothelial cells (LSECs) transplanted into the ACE of immunodeficient mice. **k** In vivo imaging of vascularization in engrafted human liver spheroids at 1-month post-tx, visualized by intravenous injection of DyLight-649-conjugated lectin (red). **l** Vasculature network (CD31, orange) and nuclei (DAPI) staining of ACE-human liver spheroid at 1-month post-tx. Max. projection image.
**a, b, g, h, i, k, l** scale bars = 100 μm. Source data are provided as a Source Data file.

innervation pattern established during the first month of engraftment was maintained thereafter (Supplementary Fig. 2c). Finally, during the engraftment process, we observed an influx of macrophages within the liver spheroids (Fig. 2i), suggesting these cells are recruited from the circulation to populate the spheroid grafts.

As humanized mouse models are powerful tools to gain insights into human-specific physiology/pathology, we explored the possibility of human liver spheroid engraftment in the mouse ACE. Thus, we generated human liver spheroids by combining primary human hepatocytes (PHH) and primary liver sinusoidal endothelial cells (LSECs) followed by transplantation into the ACE of immunodeficient NSG mice (Fig. 2j and Supplementary Fig. 2d). The LSECs served as a supporting cell type to facilitate vascularization, as spheroids formed exclusively from PHH did not become vascularized in the eye. Using this cell mixture to form human liver spheroids and transplanting them into NSG mice, we found they also achieved engraftment and vascularization by in vivo and ex vivo imaging of the graft vessels (Fig. 2k, l).

In summary, transplanted liver spheroids successfully engraft in the ACE and connect to the recipient animal by tissue remodeling.

## Engrafted spheroids retain hepatocyte features

To understand the suitability of this in vivo imaging platform to monitor hepatocyte functions, we analyzed the transcriptome of the explanted liver spheroids by RNA-seq at 2-months post-tx. First, we compared the liver spheroid explants to naïve iris samples and found hepatocyte-specific and other liver cell signature genes highly enriched in explant samples. Moreover, KEGG pathway analysis showed hits pertaining to common liver pathways (Fig. 3a). To consider the contamination of liver spheroids sample by iris tissue and to be able to more accurately compare the explant samples to in situ liver, we applied transcriptome deconvolution[13]. The results showed expression levels of genes implicated in diverse hepatic functions, with comparable levels to that of in situ liver and to the hepatocyte-enriched fraction used for spheroid formation (Fig. 3b). To corroborate the gene expression and confirm typical liver proteins, we stained ex vivo for the major glucose transporter isoform expressed in hepatocytes (Glut2) and a cell–cell adhesion protein involved in liver homeostasis (E-cadherin)[14] (Fig. 3c).

To visualize typical liver-like microstructures of the engrafted spheroids, we performed transmission electron microscopy (TEM) at 2-months post-tx (Fig. 3d and Supplementary Fig. 3). Hepatocytes within the grafts presented typical glycogen granules and characteristic cell polarity, with canalicular, lateral and sinusoidal poles[15] (Fig. 3d-2, 4). On the canalicular pole, we observed bile canaliculi characterized by microvilli and sealed on both sides by tight junctions (Fig. 3d-3, 4). The lateral poles connecting to the neighboring hepatocyte consist of tight junctions and no microvilli, while the sinusoidal

pole connects to a Disse-like space (Supplementary Fig. 3), which separates hepatocytes from the endothelium. We also found the presence of bile ductules, with luminal microvilli projections, formed by cholangiocytes, as well as macrophages in vessels and scattered within the spheroid mass (Supplementary Fig. 3).

Another typical hepatic feature is the hepatocyte functional heterogeneity along the portal-central axis, referred to as zonation and characterized by gradients of gene expression[16]. Searching our transcriptome dataset, we found the expression of typical hepatocyte zonation marker genes in engrafted spheroids (Fig. 3e). To explore the zonation of engrafted spheroids, we used RNA fluorescence in situ hybridization (FISH). We selected *Hnf4a* and *Alb* as hepatocyte identity markers and two mutually exclusive zonation marker genes *Cyp2f2* (periportal Zone 1) and *Glul* (pericentral Zone 3)[17]. The hepatocytes in engrafted liver spheroids were found to represent all three liver zones (Fig. 3f).

In summary, using transcriptomic and imaging approaches, we found that engrafted liver spheroids retain hepatocyte signatures of mature cells and demonstrate architectural similarities to the native liver.

## Hepatocyte functions can be visualized in vivo

Engrafted liver spheroids in the ACE show signatures and phenotypic features of mature hepatocytes. However, in order to serve as a platform for liver research, the engrafted hepatocytes must also retain hepatic functions which can be visualized in vivo at cellular resolution. Thus, we explored key hepatocyte functions by live imaging in the ACE following the injection of fluorescent probes (Fig. 4a).

Bile metabolism has been extensively imaged and characterized both in vitro and by intravital imaging[18,19]. The expression of multiple genes involved in bile production and xenobiotic metabolism was evident from our RNA-seq data, suggesting the presence of this system in the engrafted spheroids (Fig. 4b). Thus, we performed an in vivo bile acid export assay using 5-chloromethylfluorescein diacetate (CMFDA) which is taken up, de-esterified, excreted by hepatocytes and accumulated in bile canaliculi (Fig. 4c–e and Supplementary Movie 2). By co-injecting lectin, we differentiated the vascular from the bile canaliculi network and observed a highly organized and polarized tissue architecture (Fig. 4d), which was maintained over 6-months post-tx (Supplementary Fig. 4a). Soon after tail vein injection, CMFDA is metabolized in hepatocytes to fluorescent CMF-5 and excreted by hepatocytes. This process was visualized in the interstitial bile canaliculi, revealing a fully competent canalicular network in engrafted spheroids (Fig. 4e).

Other typical gene signatures for hepatocyte function, such as lipid metabolism (Fig. 4f), glucose and glycogen metabolism, complement and coagulation factors as well as xenobiotic

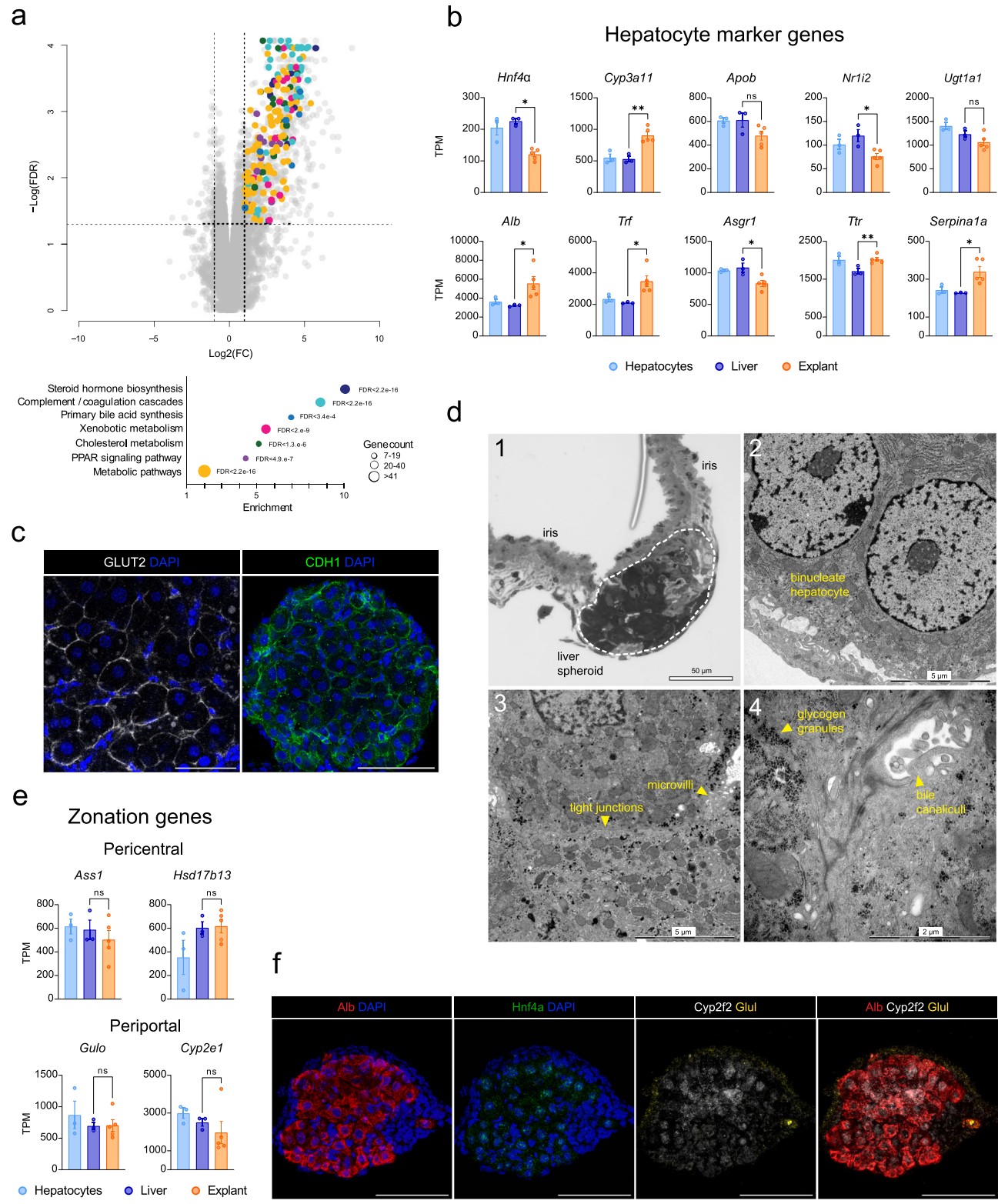

metabolism (Supplementary Fig. 4b–d), were also found in explanted spheroids. The lipid pathway and in particular the lipoprotein uptake by hepatocytes was explored by using pHrodo™ Red-LDL, which fluoresces upon uptake and incorporation into acidic endosomes. After injection of pHrodo™ Red-LDL, we observed prominent accumulation in the cytoplasm of hepatocytes and no signal in the surrounding iris (Fig. 4g). Thus, the hepatocytes within the graft express membrane LDL receptors and are active in lipoprotein uptake.

In summary, by live imaging of administered fluorescent probes, we monitored different specific hepatocyte functions in vivo showing that the hepatocytes engrafted in the ACE perform functional tasks of the in situ liver.

**Fluorescent biosensors allow longitudinal in vivo imaging**
Besides injecting fluorescent probes that reach the liver spheroids via the circulation, one can transplant liver spheroids after genetic modification in vitro to express fluorescent biosensors by, for example,

**Fig. 3 | Engrafted liver spheroids retain features of differentiated hepatocytes.** **a** Volcano plot derived from RNA-seq data at 2-months post-tx, showing gene enrichment in ACE-liver spheroids explants vs naïve iris. Color-coded genes pertain to common liver-specific pathways which are described below the volcano plot, $n = 3$ freshly isolated hepatocytes, 3 livers, 5 explants. **b** Expression levels of selected hepatocyte marker genes in explant samples compared to in situ liver at 2 months post-tx. Results shown as Transcripts Per Million (TPM), error bars indicate mean ± SEM, *$p < 0.05$, **$p < 0.01$ by $t$-test in all genes, except for *Hnf4*α, which was analyzed by Mann−Whitney test, $n = 3$ freshly isolated hepatocytes, 3 livers, 5 explants. **c** Immunofluorescence staining of hepatic markers GLUT2 and junctional protein CDH1 in ACE-liver spheroids at 2-months post-tx. Single plane images. **d** TEM images showing tissue architecture and features of ACE-liver spheroids.

Explant sample, formed of the engrafted liver spheroid and surrounding iris tissue (1); hepatocytes identified as single or double-nucleated cells, with electron-dense cytoplasm rich in mitochondria and glycogen granules (2); Bile canaliculi between neighboring hepatocytes with luminal microvilli delimited by tight junctions (3,4). **e** Expression levels of selected hepatic pericentral and periportal zonation genes in explant samples compared to in situ livers at 2-months post-tx. Results shown as TPM, error bars indicate mean ± SEM, *Gulo* and *Hsd17b13* were analyzed by $t$-test, *Ass1* and *Cyp2e1* were analyzed by Mann−Whitney test, $n = 3$ freshly isolated hepatocytes, 3 livers, 5 explants. **f** RNA in situ hybridization in sections of ACE-liver spheroids, showing transcript zonation of marker genes *Glul* (pericentral) and *Cyp2f2* (periportal), and *Alb* and *Hnf4*α to identify hepatocyte cells. **c, f** scale bars = 100 μm. Source data are provided as a Source Data file.

viral vectors (Supplementary Fig. 5a). We transduced cells during spheroid formation with adeno-associated virus (AAV8, AAV9) encoding GFP and longitudinally monitored their permanence post-tx for up to 4-months (Supplementary Fig. 5b, c). Similarly, liver spheroids originating from transgenic mouse models with endogenous expression of fluorescent biosensors can be used to address specific research questions. Here, as proof-of-principle, we extracted primary hepatocytes and generated spheroids from B6.Cg-Tg(Fucci)504/596Bsi (FUCCI) transgenic mice (Fig. 5a, b), which allow monitoring of cell cycle progression by color-coding cells in G0/G1 phases (red) and S/G2/M phases (green)[20]. After validation by inducing proliferation of the mouse liver spheroids in vitro (Supplementary Fig. 5d), FUCCI-expressing liver spheroids were transplanted into the ACE and imaged longitudinally in vivo (Fig. 5c, d). At 2-days post-tx, in vivo imaging showed a boost in cell cycle activity similar to what is observed in the regenerating liver[21], in comparison to the absence of cell cycle activity seen during in vitro culture (Fig. 5c, d and Supplementary Fig. 5d). At 1-week post-tx, liver spheroids returned to lower levels of cell cycle activity and maintained this low but constant DNA replication throughout the following weeks. During this time, we were able to visualize distinct stages of the cell cycle in transplanted liver spheroids at single-cell resolution (Fig. 5c, e). Additionally, ex vivo immunofluorescence staining for the proliferation marker Ki67 showed evidence of cell cycle activity in hepatocytes, corroborating the in vivo observations (Fig. 5f).

In summary, we showcase the longitudinal in vivo imaging of liver spheroids in the eye, either originated from transgenic reporter mice or in vitro genetically modified liver cells to express fluorescent proteins.

## ACE-spheroids report on diet-induced lipid accumulation

To evaluate the potential of this platform in acting as a reporter of in situ liver functions, we tested whether the ACE-liver spheroids would respond to diet-induced hepatic lipid accumulation in a similar way as in situ hepatocytes of the recipient mice. Feeding mice a high-fat-high-fructose diet (HFHFrD) causes obesity, insulin resistance, and liver lipid accumulation[22,23]. Mice with intraocular liver spheroids were fed an HFHFrD for 12 weeks to compare lipid accumulation in ACE-liver spheroids and in situ liver (Fig. 6a, b). Ex vivo immunostaining using Nile Red showed that ACE-liver spheroids of HFHFrD-fed recipient animals accumulate lipid droplets within hepatocytes in similar amounts and size, as hepatocytes in the endogenous liver (Fig. 6c−e). We further confirmed the formation of lipid droplets in engrafted hepatocytes by TEM (Fig. 6f). Most importantly, by injecting a recently developed fluorescent lipid dye, namely SF44[24], we visualized the accumulation of lipid droplets at subcellular resolution in vivo (Fig. 6g, h).

In summary, we showed the potential of using ACE-liver spheroids as reporters of endogenous liver function. Here, engrafted liver spheroids sensed metabolic changes in the recipient body caused by HFHFrD, mirroring endogenous hepatosteatosis.

## Discussion

Research focused on understanding organ biology in health and disease to a major extent relies on the visualization of cellular processes. Within liver research, non-invasive longitudinal in vivo imaging with cellular resolution is currently lacking. In vitro studies using 3D spheroids assembled from primary hepatocytes have facilitated important discoveries over the past years. Although in vitro culture allows scalable research, this system lacks physiological stimuli and cultured spheroids have a limited lifetime of around 1 month[25]. Real-time and in situ imagery in living animals has been achieved with invasive and terminal procedures (intravital imaging)[26] or through abdominal windows, which entails complex surgery and after-care[27]. Here, we describe a strategy to overcome this bottleneck by transplanting liver spheroids into the anterior chamber of the mouse eye (ACE), allowing longitudinal, non-invasive monitoring of liver-like functions at cellular resolution. Our laboratory pioneered the ACE-imaging approach using pancreatic islets, which is currently exploited for diabetes research[11,12]. However, liver tissue has not been successfully transplanted for in vivo imaging so far, even though the first attempt of transplanting it into the ACE was tested already in the 1930s[28]. Here, we showcase the many possibilities this platform opens up for liver research, in academia as well as in pharma-biotech fields.

Following transplantation into the ACE, we show that liver spheroids become vascularized and innervated, thus providing access to the complex composition of blood and nervous input. Moreover, despite the presence of both sympathetic and parasympathetic innervation in the surrounding iris, we found that ACE-liver spheroids become innervated with only sympathetic nerves, resembling the endogenous liver[29,30]. These findings suggest that parenchymal cells within the liver spheroid grafts dictate the features of vessel and nerve inputs, which determine their correct functioning.

One pivotal feature for this platform to study hepatocyte functions in the ACE is the proof that they retain signatures of the in situ liver. We studied this by (1) analyzing the transcriptome of the engrafted spheroids and comparing it with that of the endogenous liver, (2) ex vivo imaging techniques, and, most importantly, (3) testing specialized hepatocyte functions in vivo. We were able to apply imaging fluorescent probes available on the market and show vascularization, blood flow, bile acid transport, LDL uptake, and lipid droplet accumulation. Interestingly, the timeframe and dynamics of bile acid export were similar to that of the in situ liver[31,32]. Importantly, what we have shown here could be replicated with any other injectable fluorescent probe to report on specific liver-related research questions. Similarly, we showed that cells equipped with fluorescent biosensors within the transplanted spheroids allow longitudinal monitoring of the reporter of interest. As proof-of-concept, we used the cell cycle indicator FUCCI, which can be used in future studies as a tool to study liver regeneration dynamics. Fluorescent probes or biosensors will also enable studies in disease models, where they can act as reporters of in situ liver functions. Lipid accumulation is a feature of metabolic-associated steatotic liver disease (MASLD), including metabolic dysfunction-associated steatohepatitis

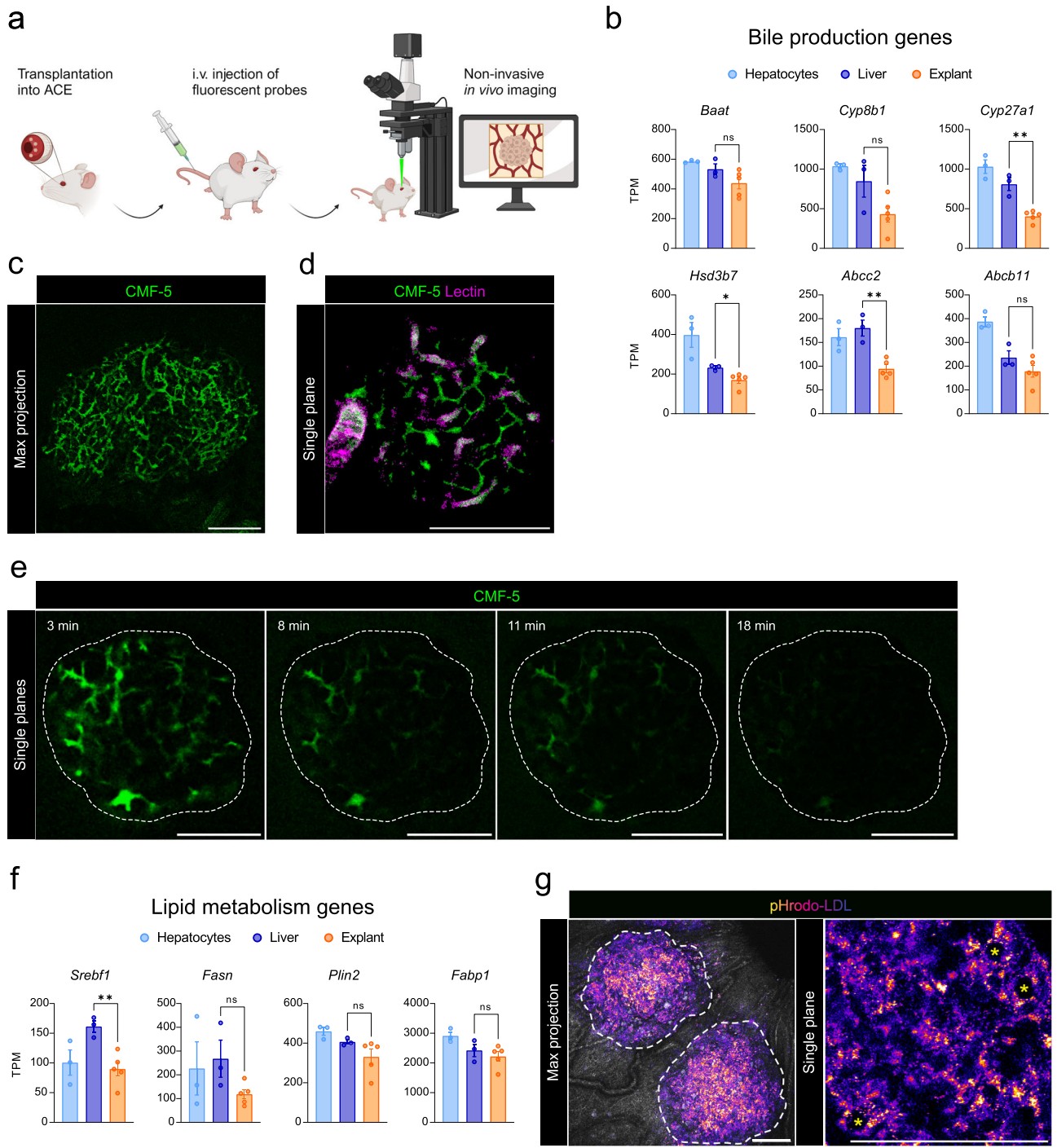

**Fig. 4 | Hepatocyte functions can be visualized in vivo by the administration of fluorescent probes. a** Experimental design of in vivo imaging: after 1 month of engraftment, graft-bearing mice were injected with fluorescent probes (CMFDA, pHrodo™-LDL), and hepatocyte functionality was assessed by non-invasive in vivo imaging. **b** Expression levels of selected bile production genes in explant samples compared to in situ livers at 2-months post-tx. Results shown as TPM, error bars indicate mean ± SEM, *$p < 0.05$, **$p < 0.01$ by $t$-test in all genes, except for *Abcb11*, which was analyzed by Mann–Whitney test, $n = 3$ freshly isolated hepatocytes, 3 livers, 5 explants. **c** CMF-5 (green) fluorescent signal collected during in vivo imaging reveals an intricate bile canaliculi network within the ACE-liver spheroid. **d** Co-injection of CMFDA (green) with lectin to mark vessels (magenta) during in vivo imaging. **e** Time series of in vivo imaging in ACE-liver spheroids, showing the excretion of CMF-5 (green) by hepatocytes and its accumulation in bile canaliculi. **f** Expression levels of selected lipid metabolism genes in explant samples compared to in situ livers at 2-months post-tx. Results shown as TPM, error bars indicate mean ± SEM, *p < 0.05, **$p < 0.01$ by $t$-test in all genes, $n = 3$ freshly isolated hepatocytes, 3 livers, 5 explants. **g** pHrodo™-LDL fluorescent signal (purple scale) collected during in vivo imaging showing accumulation of the probe within the cytoplasm of hepatocytes in ACE-liver spheroids (nuclei are marked by asterisks). **c**, **d**, **e**, **g** scale bars = 100 μm. Source data are provided as a Source Data file.

(MASH), the prevalence of which is exponentially increasing especially in Western countries[33]. After diet intervention to induce liver steatosis, we show that engrafted liver spheroids accumulate lipids mirroring the in situ liver, suggesting that this platform can be used to monitor disease development. These ACE-imaging studies can be complemented by additional physiological parameters obtained longitudinally from the same animals such as insulin and glucose tolerance, omics on serum as well as aqueous humor samples.

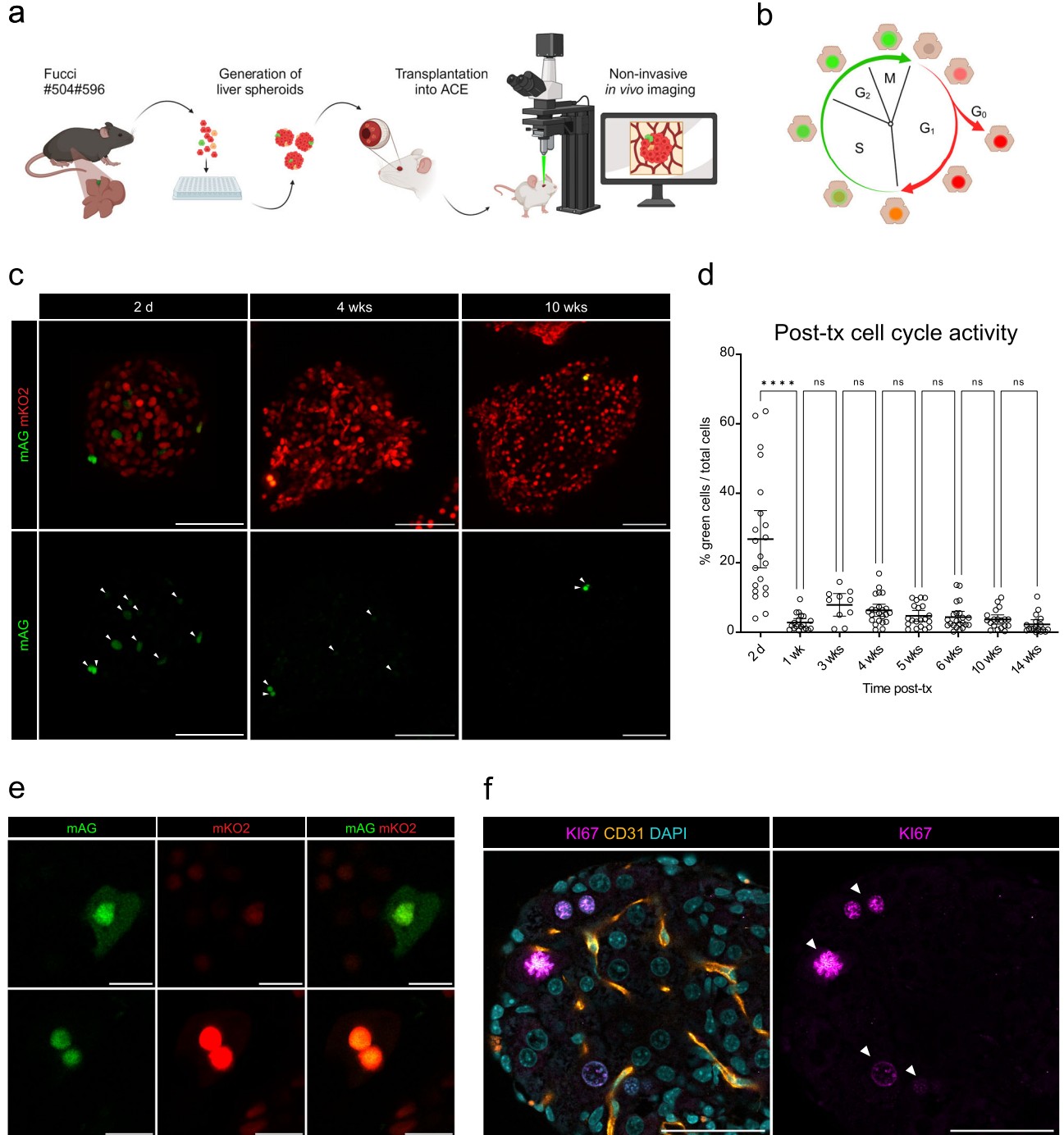

**Fig. 5 | Engrafted spheroids expressing fluorescent biosensors allow longitudinal in vivo imaging. a** Diagram of experimental design: liver spheroids were generated from primary liver cells (enriched for hepatocytes) from FUCCI reporter mice and transplanted into the ACE for longitudinal in vivo imaging. **b** Scheme of the FUCCI biosensor which reports on cell cycle progression by color-coding cells in G0/G1 phases (red) and S/G2/M phases (green). **c** In vivo longitudinal imaging of different FUCCI-spheroids at different time-points post-tx, showing cells in different stages of the cell cycle (S/G2/M cells in green and G1/G0 cells in red). White arrowheads indicate green cells. Scale bar = 200 μm.

**d** Quantification of the percentage of S/G2/M cells in ACE-FUCCI-liver spheroids post-tx. Results are shown as mean with 95% CI. ****$p < 0.0001$ by Kruskal–Wallis test, $n = 10$-23 spheroids in 3 recipient mice. **e** In vivo imaging captures different cell cycle stages of the FUCCI hepatocytes in the ACE at single-cell resolution: cell in G2/M phase (top) and cell in transition between G1 and S phase (bottom). Scale bar = 20 μm. **f** Immunofluorescence staining of whole-mount ACE-liver spheroid and surrounding iris tissue, showing Ki67-positive hepatocytes (magenta) and vessels (CD31, orange) at 2-months post-tx. Single plane image, scale bar = 50 μm. Source data are provided as a Source Data file.

In this work, we have focused on the transplantation of mouse liver spheroids into recipient animals. However, we also demonstrate the possibility of engraftment of human liver spheroids in the ACE of immunodeficient mice, given that humanized mouse models are highly valuable research and screening tools. Of note, when transplanting liver spheroids made of purified primary human hepatocytes

(PHH), these spheroids failed to engraft and become vascularized in the mouse eye. We attribute this to the purchased PHH being highly purified and therefore lacking supporting cell types, such as endothelial cells or Kupffer cells, both of which are known to secrete proangiogenic factors and growth factors, which promote vascularization[34,35]. Additionally, there could be species-specific

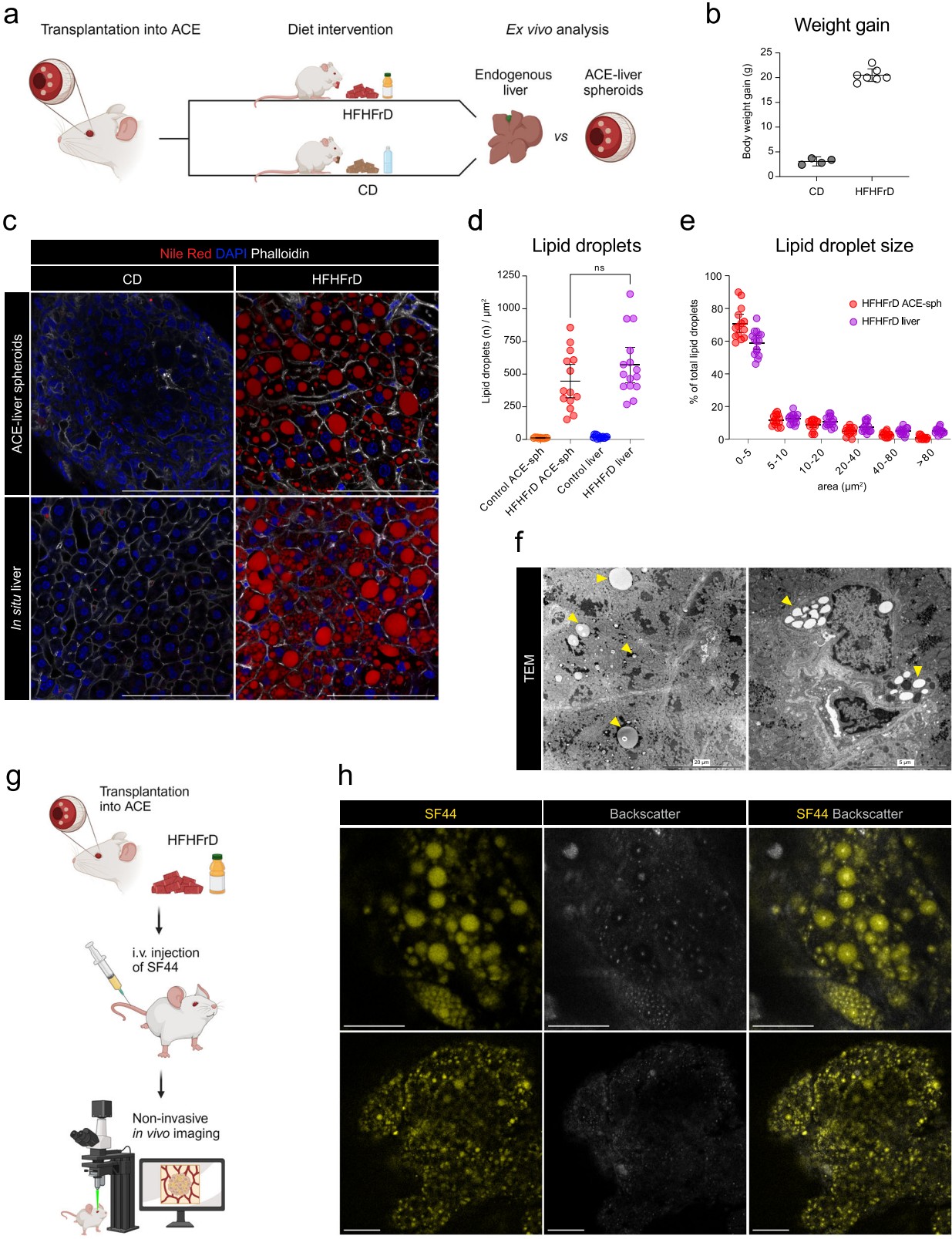

endocrine signals that are hindered by the transplantation of human tissue into mouse recipients. Therefore, we show that the addition of primary human LSECs facilitates the vascularization of the graft, a strategy that has been used by others. In a study by Takebe et al., iPSC-derived hepatocytes were cultured with supportive mesenchymal and endothelial cells to form 3D-liver buds, which were then transplanted onto the mouse dura mater where they successfully engrafted and became vascularized[36].

The current state-of-the-art method to achieve a humanized liver model is through the injection of human hepatocytes via the portal vein to form a chimeric liver[37]. Our technological advance would not allow the replacement or functional substitution of the endogenous

**Fig. 6 | ACE-liver spheroids report on the endogenous diet-induced lipid accumulation of in situ liver. a** Diagram of experimental design: graft-bearing mice were fed a high-fat-high-fructose (HFHFrD) or control (CD) diet for 12 weeks and quantification of lipid droplets was performed ex vivo on ACE-liver spheroids and in situ liver samples. **b** Body weight gain in recipient mice at 12 weeks of HFHFrD or CD intervention, error bars indicate mean ± SD, $n = 4–7$ mice per group. **c** Immunofluorescence staining of whole-mount ACE-liver spheroids and in situ liver, showing lipid droplets (Nile Red, red) within hepatocytes, delineated by F-actin (Phalloidin, white). Single plane images, scale bar = 100 μm. **d** Amount of lipid droplets per image plain in ACE-liver spheroids and in situ liver samples of HFHFrD and CD-fed recipient mice. Results shown as mean with 95% CI by One-way ANOVA test, $n = 11–15$ planes per spheroid, 3 spheroids/livers in 3 recipient mice. **e** Lipid droplet size distribution per image plane comparison between HFHFrD-fed ACE-liver spheroids and in situ liver. Results are shown as mean with 95% CI, $n = 14–15$ planes per spheroid, 3 spheroids/livers in 3 recipient mice. **f** TEM images showing lipid droplets (yellow arrowheads) within hepatocytes (left) and other intra-spheroid cell types (right). **g** Diagram of experimental design: transplanted mice were fed an HFHFrD or CD for 12 weeks and lipid droplets were visualized by i.v. injection of SF44 (fluorescent lipid dye) during in vivo imaging. **h** In vivo imaging of lipid droplets (SF44, yellow) and backscatter signal (white) within engrafted liver spheroids. Single plane images, scale bar = 50 μm. Source data are provided as a Source Data file.

liver, but it is highly appealing as a monitoring platform or drug validation tool. We foresee that the ACE-imaging platform can pose as a platform for pre-clinical testing of human therapeutics, with the advantages of being a much simpler and attainable approach compared to current model systems such as humanized liver model[38] or installation of abdominal body windows in mice[27]. We think this platform would be highly applicable in combination with other high-throughput techniques. For example, as a platform for in vivo validation of compounds previously tested in in vitro 3D-liver models, which cannot replicate the in vivo milieu. Probes/biosensors reporting on hepatotoxicity of ACE liver spheroids will be a unique tool for studying drug efficacy and safety in vivo. This platform will also have the potential for a precision medicine approach, by transplantation of patient-derived liver spheroids or stem cell-derived liver organoids[39] to test their specific reactions to drugs or other therapies.

The ACE-liver spheroid imaging platform has its limitations and engrafted liver spheroids as reporters of the liver in situ should be accurately studied and validated depending on the research aim and readout. ACE-liver spheroids lack portal vein input and the complex hepatic lobule architecture, among others, and thus should not be considered to represent bona fide liver. However, bearing in mind these considerations, hepatocyte functions are retained in the spheroids, making this platform an invaluable tool for answering specific biological questions.

In conclusion, we have established and characterized a platform for in vivo imaging of mouse liver spheroids, in which the graft can be imaged non-invasively and repeatedly at different time-points (longitudinally) in the same animal at single-cell resolution. The liver spheroids preserve liver-like features and retain hepatic differentiation and functionality. By different proof-of-concept experiments, we have shown the monitoring capabilities of this platform in different areas of liver research, such as metabolic disease and liver regeneration. We believe this platform will be of great value in both basic research and translational studies, within the limitations dictated by the latest international directives on the use of animal models.

## Methods

### Ethical authorization and justification

All animal experiments were performed in accordance with the Animal Experiment Ethics Committee at Karolinska Institutet. This work falls under the broader ethical permission approved by Jordbruksverket: "Studies on the function of hormone-releasing and hormone-stimulated cells and related cells/tissues in normal and diabetic animal models and in transplanted tissues and cells" (n. 6362-2023, previous 16454-2022, 17431-2021, 8822-2020). The utilization of our in vivo imaging platform directly addresses two of the 3R principles "Reduction" and "Refinement" in compliance with the European Directive 2010/63/EU.

### Animal welfare, husbandry, behavior and sacrifice

Experiments were conducted on both male and female adult mice (≥3 months old) housed in groups of 2–6 animals at 22 °C and 40–55% humidity, on a 12/12-h dark/light cycle with ad libitum access to normal chow diet (R70, Lantmännen, Stockholm, Sweden) and water, unless otherwise stated. The following mouse strains were used: B6(Cg)-Tyrc-2J/J mice (Albino B6/J mice) and NOD.Cg-Prkdcscid Il2rgtm1Wjl/SzJ (NSG) mice were purchased from the Jackson Laboratory and bred in-house; B6.Cg-Tg(Fucci)504Bsi Tg(Fucci)596Bsi (FUCCI) mice were provided by the Bergmann group (Karolinska Institutet, Stockholm, Sweden). After transplantation of the liver spheroids in the ACE, the animals recover quickly from the surgical intervention, with no need for post-operative care, and the transplantation does not affect their behavior. We considered normal behavior to be aspects such as no increased fighting between males, normal feeding and growth, normal levels of activity, and nest-making. We have monitored weight gain/loss and non-fasting blood glucose levels in transplanted and non-transplanted animals, as an indirect measure of sustained stress and metabolic alterations. We found that neither weight nor blood glucose levels significantly differed between the two groups over a period of 1-month post-transplantation (Supplementary Fig. 6). Additionally, we performed in vivo imaging at 3-weeks post-transplantation in recipient animals, which did not have a negative effect on their health. Moreover, transplanted animals have been maintained for over 1 year, with no apparent effect on their health or longevity. The mice always remained group-housed, with no need for isolation neither before nor after surgery nor in vivo imaging. The sacrifice of the animals was carried out by cervical dislocation at the end of the experiment and no health deterioration of the animal was linked to the transplantation surgery or in vivo imaging sessions.

### Diet intervention

For high-fat high-fructose diet (HFHFrD) intervention, male mice were fed ad libitum with a high-fat diet (60% kcal from fat, TD.06414, Envigo, Indianapolis, US) and 32% w/v of Fructose-D (Sigma-Aldrich, Massachusetts, US) in tap water.

### Isolation of mouse primary liver cells and generation of mouse liver spheroids

Mouse primary liver cells enriched in hepatocytes were isolated from mouse liver following a two-step collagenase perfusion protocol. Briefly, the liver was cannulated using a 27-G needle through the inferior vena cava and perfused using a peristaltic pump and liquids heated in a 42 °C water bath. Approximately 20 ml of perfusion buffer (HBSS w/o Ca²⁺/Mg²⁺, HEPES 25 mM, EDTA 0.5 mM) was perfused at a rate of 2.5 ml/min and the portal vein was immediately severed to allow the passage of solutions through the organ. Once the liver was drained and cleared of blood, approximately 12 ml of digestion buffer (25 μg/ml Liberase™ TM Research Grade (Sigma-Aldrich) in William's E (Gibco, Thermo Fisher, Massachusetts, US) supplemented with 1x Penicillin-Streptomycin (Gibco)) was perfused at 2.5 ml/min and the tissue was checked for signs of digestion. The digested liver was transferred to a 10 cm petri dish containing 10 ml of cold mouse plating medium (William's E, 5% FBS, Insulin-Transferrin-Selenium (Gibco) 1x, Dexamethasone 1 μM, L-Glutamine 1%, Penicillin-Streptomycin 1%), the gall bladder was discarded and the liver cells were released from the connective tissue using a cell lifter. The cell slurry was strained through

a 70 μm filter into a 50 ml Falcon tube and spun at 50 x $g$ for 5 min. The supernatant was removed and the cells were resuspended in 10 ml of cold plating medium. A mixture of 9 ml of Percoll (Invitrogen, Thermo Fisher, Massachusetts, US) and 1 ml of 10x PBS was added to the cell suspension and the tube was inverted 10 times. The cells were spun at 200 x $g$ for 10 min to form a gradient. The hepatocyte-enriched phase formed a pellet, while dead hepatocytes and non-parenchymal cells floated in the gradient. The live hepatocyte fraction was resuspended in 20 ml of cold plating media and spun at 50 x $g$ for 5 min. The cells were tested for viability and counted manually using Trypan blue. The hepatocytes were seeded in plating medium at 1200 cells/well into 96-well ultra-low adherence plates (Nunclon™ Sphera™ 96-Well, U-Shaped-Bottom Microplate, Thermo Fisher) and the plates were spun at 200 x $g$ for 3 min. The cells were left to form spheroids for 5 days, at which stage 50% of the medium was replaced with serum-free mouse maintenance medium (William's E, Insulin-Transferrin-Selenium (Gibco) 1x, Dexamethasone 100 nM, L-Glutamine 1%, Penicillin-Streptomycin 1%). This step was repeated every 48 h up to day 10.

### Size measurement of liver spheroids
Liver spheroids both in vitro and in vivo were measured by calculating the average of vertical and horizontal diameters. In vitro, spheroids were imaged in 96-well plates using brightfield microscopy and in vivo, spheroids were imaged using light backscatter (exciting and detecting light at 633 nm) to differentiate the spheroid mass from the iris.

### Generation of human liver spheroids
Cryopreserved purified primary human hepatocytes (PHH) and liver sinusoidal endothelial cells (LSECs) were obtained from suppliers BioIVT (New York, US) and Lonza (Basel, Switzerland), respectively (Supplementary Table 1). Ethical approval was therefore not necessary for the generation of human liver spheroids. PHHs and LSECs were mixed and seeded at a 2:1 ratio into BIOFLOAT™ ultra-low adherence 96-well plates (FaCellitate, Mannheim, Germany) in human plating medium (William's E (Gibco), 10% FBS, Dexamethasone 100 nM, Penicillin 1%, L-Glutamine 2 mM, Insulin (Gibco) 10 μg/ml, Transferrin (Gibco) 5.5 μg/ml, and Sodium Selenite (Gibco) 6.7 ng/ml). The plates were spun at 200 x $g$ for 3 min. The spheroids were allowed to aggregate and after 5 days, 50% of the plating medium was replaced with fresh media every 48 h, up to day 10.

### Transplantation of liver spheroids into the anterior chamber of the eye (ACE)
Liver spheroids were transplanted at day 10 of formation into the ACE of recipient mice. The mice were anaesthetized using isoflurane (Baxter, Illinois, US) delivered by the Univentor 400 anesthesia unit (Univentor, Zejtun, Malta), and the eye was positioned and secured for surgery using a custom-built stereotaxic head holder (Fig. 1c). Under a stereo microscope (M80, Leica, Wetzlar, Germany), the cornea was punctured at one side of the pupil with a 23-G needle. The liver spheroids were aspirated in their culture medium into a custom-made blunt glass cannula connected to a 0.5 ml Hamilton syringe (Hamilton Company, Nevada, US) via 0.4-mm polythene tubing. The cannula was inserted into the puncture hole and the liver spheroids were released into the ACE, where they settled onto the iris. Both eyes were transplanted with 5–10 spheroids per eye. The operated eyes were doused in Oculentum simplex (APL, Stockholm, Sweden) eye ointment to help with corneal abrasion and inflammation. Post-operative analgesic Temgesic (2 μl/g mouse at 0.05 mg/ml in sterile saline, Indivior, Virginia, US) was administered by subcutaneous injection approximately 5 min before awakening. The mice were placed on a heating pad until fully recovered from the anesthesia and showed no post-operative signs of pain or irritation in the operated eyes.

### In vitro viral transduction of mouse liver spheroids
Mouse primary liver cells enriched in hepatocytes were seeded in plating medium containing 1 μl/ml of adeno-associated virus (AAV), namely AAV8-CAG-GFP or AAV9-CAG-GFP (Charles River, Massachusetts, US). The cells were left to aggregate for 5 days, at which stage 50% of the medium was replaced with serum-free mouse maintenance media. This step was repeated every 48 h up to day 10.

### In vitro cell viability assay
Live mouse liver spheroids were incubated with Live-or-Dye NucFix™ Red (Biotium, California, US) for 30 min at RT, washed in PBS and then fixed in neutral buffered formalin (NBF, Sigma-Aldrich) 10% for 1 h at RT. The spheroids were stained for DAPI and mounted into iSpacer wells (Double Sided Sticky, 0.25 mm, SUNJin Lab, Hsinchu City, Taiwan) on microscope glass slides in tissue-clearing solution RapiClear 1.47 (SUNJin Lab). The spheroids were imaged as $Z$-stacks of 4 μm step-size, with excitation 520 and emission 540-600 nm. Red (dead) cells throughout the $Z$-stack were counted manually.

### Microscopy set up for intraocular in vivo imaging of liver spheroid grafts
Mice were anaesthetized with isoflurane and placed on a heating pad. A custom-built stereotaxic head holder allowed positioning and fixture of the mouse eye toward the objective (Fig. 1c). For imaging, an upright laser scanning confocal microscope (TCS SP5 II, Leica) equipped with a long-distance water-dipping objective (HCX IRAPO L 25x/0.95W, Leica) was used. Viscotears (Novartis, Basel, Switzerland) were used as the immersion liquid between the eye and the objective. At the end of the imaging sessions, the imaged eyes were treated with Oculentum simplex eye ointment to lubricate the cornea and help cure any possible abrasions or inflammation resulting from the eye manipulation. We detailed the microscopy imaging settings used for the different fluorescent probes and reporter proteins in Supplementary Table 2. Following these specifications, the grafts can be imaged safely, over multiple imaging sessions, with no photodamage to the cells.

### In vivo imaging
**Vessels.** For visualization of blood vessels in the ACE, Lycopersicon Esculentum Lectin conjugated to DyLight-649 (excitation 633 and emission 650−700 nm, Invitrogen) or FITC (excitation 488 and emission 510−540 nm, Sigma-Aldrich) was administered via intravenous tail vein injection (100 μl/mouse at 1 mg/ml). The structure of the iris and spheroids was imaged using light backscatter, by exciting and detecting light at 561 nm. The liver spheroids were imaged as $Z$-stacks of 4 μm step-size.

**RBC labeling.** For in vivo imaging of labeled erythrocytes, 100 μl of mouse blood was mixed with 200 μl of blood plasma buffer (BPB) (NaCl 128 mM, D-Glucose 15 mM, HEPES 10 mM, NaHCO$_3$ 4.2 mM, KCl 3 mM, MgCl$_2$ 2 mM, KH$_2$PO$_4$ 1 mM in ddH$_2$O, pH 7.4) and centrifuged at 250 x $g$ for 5 min. The supernatant was discarded and the red blood cell (RBC) pellet was resuspended in 400 μl of BPB. A mixture of 400 μl of Diluent C (Sigma-Aldrich) and 10 μl of DiD dye (Invitrogen) was added to the resuspended RBCs. The cells were incubated at 37 °C for 10 min and periodically mixed. Thereafter, 200 μl of mouse serum (Invitrogen) was added and cells were incubated for an additional 1 min at 37 °C. The cells were then centrifuged at 250 x $g$ for 5 min and washed twice by resuspending in 1 ml of BPB containing 10% mouse serum. After the final wash, the unbound dye was removed and the labeled RBCs were resuspended in 300 μl of BPB. Prior to in vivo imaging, 100 μl/mouse of labeled RBCs were injected via the tail vein, together with 100 μl/mouse of FITC-conjugated Lectin. The circulating RBCs were imaged with excitation at 633 and emission at 650−700 nm.

**Bile canaliculi.** To visualize the bile canaliculi network in engrafted liver spheroids, Green CMFDA (reconstituted in DMSO, Abcam, Cambridge, UK) 100 µg/mouse in PBS 10% FBS, was administered via intravenous tail vein injection. In vivo recordings were made prior to injection and at specified time intervals thereafter. CMF-5 signal was recorded in single planes with excitation 488 and emission 510–540 nm.

**LDL uptake.** To observe the uptake of LDL by engrafted liver spheroids, pHrodo™ Red-LDL (100 µl/mouse at 1 mg/ml, Invitrogen) was injected in the tail vein. In vivo imaging was performed prior to injection and after 1 h, to allow accumulation of the labeled LDL within cells. PHrodo™ Red-LDL signal was detectable with excitation 561 and emission 570-620 nm. The structure of the iris and spheroids was imaged using light backscatter, by exciting and detecting light at 633 nm.

**FUCCI cell cycle indicator.** Liver spheroids were generated from FUCCI mouse primary liver cells enriched in hepatocytes and transplanted into the ACE. The liver spheroids were imaged in vivo at 2 days, 1, 3, 6, 10, and 14 weeks post-transplantation. The two fluorophores were imaged in vivo simultaneously using the 488 nm laser to excite both proteins and detect mAG at emission 490–530 nm and mKO2 at emission 560-580 nm. For in vitro proliferation experiments, FUCCI-liver spheroids were treated in vitro with YAC cocktail[40] (10 µM ROCK inhibitor Y-27632, 5 µM TGFB inhibitor A-8301, 3 µM GSK3 inhibitor CHIR99021, 10 ng/ml recombinant EGF, 40 ng/ml recombinant HGF in mouse maintenance media) for 72 h to induce proliferation.

**Lipid droplets.** Seoul Fluor 44 (SF44, SPARK Biopharma, Seoul, Korea) powder was reconstituted in DMSO and further diluted to a solution of 5.8 mM in PBS 10% FBS which was administered by tail vein injection (100 µl/mouse). Lipid droplets were imaged approximately 30 min post-injection and SF44 was detected with excitation 488 and emission 580–700 nm.

### Immunofluorescence staining
Whole eyes were fixed in 10% NBF for 3 h at RT, followed by two washes with 1x PBS. The eyes were manually dissected to obtain the engrafted liver spheroids, embedded in the surrounding iris tissue. Briefly, the eyeball was divided by cutting along the ciliary body to separate the ACE from the posterior and vitreous chambers. The lens was removed from behind the iris and the remaining cornea and iris were cut into two semicircles. Gently, the iris was released from its connection with the remaining ciliary body and the cornea. The explants, i.e. the engrafted liver spheroids and supporting iris tissue, were incubated in blocking and permeabilization solution (2% Triton X-100, 1x PowerBlock (Biogenex, California, US) and 10% FBS in 1x PBS) overnight at 4 °C. Both primary and secondary antibody incubations were performed in 1% Triton X-100, 1x PowerBlock, and 10% FBS in 1x PBS. Primary antibody incubation was carried out overnight at 4 °C. The primary antibodies and dilutions used are reported in Supplementary Table 3. Alexa Fluor-conjugated secondary antibodies Phalloidin-647 (Thermo Fisher, 1:200), and Nile Red (Thermo Fisher at 1 µl/ml) were incubated for 2 h at RT together with DAPI (Abcam at 1 µl/ml). Finally, the explants were mounted into iSpacer wells on microscope glass slides in tissue-clearing solution (Rapi-Clear 1.47) and imaged on a confocal laser scanning confocal microscope (TCS SP8, Leica).

### Bulk RNA-sequencing
Eyes from transplanted animals were taken at 2 months post-tx and were manually dissected in cold PBS to obtain the explants, formed of engrafted liver spheroids and surrounding iris tissue. The explant samples were processed for bulk RNA-seq, performed by the Single-Cell Core Facility for the Flemingsberg campus (SICOF, Stockholm, Sweden). Differential gene expression analyses between liver spheroid explants and naïve iris were carried out using the Limma package[41] (Bioconductor). The expression levels of candidate genes in liver spheroid explants were calculated using CIBERSORT[13], which estimates cell compositions in mixed tissues based on gene expression profiles of individual compartments. In brief, transcript levels in naïve iris samples were subtracted from normalized counts (transcripts per million (TPM)) of liver spheroid explant samples, applying the following formula, where $F_i$ is the fraction of iris tissue in the explant sample as calculated by CIBERSORT, based on iris and liver transcriptomic signatures:

$$\text{Explant normalized TPM} = \frac{\text{TPM}_{\text{raw explant}} - (\overline{\text{TPM}_{\text{iris-only}}} * F_i)}{1 - F_i}$$

### Transmission electron microscopy (TEM)
Whole eyes were dissected in cold PBS to obtain the engrafted liver spheroids and the surrounding iris. The fresh explant was fixed in 2.5% glutaraldehyde and 1% formaldehyde in 0.1 M phosphate buffer pH 7.4 for 1 h at RT followed by storage at 4 °C until further processing. After fixation, the explant was rinsed in 0.1 M phosphate buffer and treated with 2% osmium tetroxide in 0.1 M phosphate buffer pH 7.4 for 2 h at 4 °C. Following stepwise dehydration in ethanol and acetone, the graft was embedded in LX-112 resin (Ladd, Vermont, US). Ultrathin sections (approximately 80–100 nm) were prepared using an EM UC7 (Leica), contrasted with uranyl acetate, followed by lead citrate, and finally examined in an HT7700 transmission electron microscope (Hitachi High-Technologies, Tokyo, Japan) at 80 kV. Digital images were acquired using a 2kx2k Veleta CCD camera (Olympus Soft Imaging Solutions, Münster, Germany).

### Fluorescence RNA in situ hybridization
Whole eyes were fixed in 10% NBF for 12 h at RT and then manually dissected to obtain the engrafted liver spheroids, along with the supporting iris. Explant pieces were encapsulated in HistoGel™ (EprediaTM, Thermo Fisher) and placed in CellSafe+ Biopsy Capsules (CellPath, Newtown, UK) and submerged in 70% ethanol until processed using Tissue Tek VIP (Sakura Finetek, Alphen aan den Rijn, The Netherlands). The treated explants were embedded in paraffin and cut into 4 µm sections using a microtome (Microm HM360, Thermo Fisher). The sections were used for RNAscope™ Multiplex Fluorescent V2 Assay (ACDBio Newark, US), following the company's protocol. Images were collected in Z-stacks on a confocal microscope (TCS SP8, Leica).

### Image analysis and quantification
All image analysis and quantification were performed using ImageJ (Maryland, US) unless otherwise stated.

**Vessel analysis.** Z-stacks acquired during in vivo imaging were converted to maximum projections, adjusted for arbitrary threshold, and converted to binary images. The binary images were processed by dilate and erode functions and the spheroid surface was traced manually using the drawing tool. The mean gray value and area measurements were used to calculate the percentage of vessel area over the total spheroid area. Vessel branch points were counted manually on maximum projection images. Vessel diameters were measured manually on the Leica LAS X Software.

**FUCCI spheroid analysis.** Z-stacks acquired during in vivo imaging were analyzed using Imaris (Zurich, Switzerland). The mAG and mKO2 positive cells were counted using the Spots function and setting the

approximate spot diameter to 8 μm (size of hepatocyte nucleus) and a manual threshold was maintained throughout the analysis.

**Lipid droplet analysis.** Single plane images were adjusted for threshold (Yen mode) and made into binary images. The images were processed by the functions: dilate, erode, fill holes, and watershed. Images were automatically quantified using Analyze particles (pixel units, 2-infinity) and the number of particles and area were recorded.

## Statistical analysis and reproducibility

All statistical tests along with sample sizes for each test were specified in the figure legends and were performed using GraphPad Prism 9.5.1. Differences with $p$-value < 0.05 were considered statistically significant. Two-sided Student $t$-tests or Mann–Whitney were applied for statistical analysis between two groups depending on data distribution. One-way ANOVA (Tukey's post hoc test), Two-way ANOVA, or Kruskal–Wallis (Dunn's post hoc test) tests were applied for statistical analysis between three or more groups depending on data distribution. All data sets include at least three biological replicates per experimental group.

## Reporting summary

Further information on research design is available in the Nature Portfolio Reporting Summary linked to this article.

## Data availability

The RNA-seq dataset for sequencing analysis generated in this study has been deposited in the GEO database under the accession code GSE245944. Source data used to generate the graphs in the figures are provided in this paper. The rest of the data that support this study are available in the manuscript and supplementary information or from the corresponding author upon request. Source data are provided in this paper.

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

## Acknowledgements

F.L.-B., B.L., I.B.L., N.M. and P.-O.B. were supported by the Swedish Diabetes Association, Funds of Karolinska Institutet, The Swedish Research Council, Novo Nordisk Foundation, The Family Erling-Persson Foundation, Strategic Research Program in Diabetes at Karolinska Institutet, The Family Knut and Alice Wallenberg Foundation, The Jonas & Christina af Jochnick Foundation, Swedish Association for Diabetology and ERC-2018-AdG 834860-EYELETS. N.O.V. and V.M.L. were supported by the Swedish Research Council, the Ruth och Richard Julins Foundation for Gastroenterology, Knut and Alice Wallenberg Foundation, International Foundation for Ethical Research, and the Robert Bosch Foundation, Stuttgart, Germany. M.B. and O.B. were supported by the Center for Regenerative Therapies Dresden, the Karolinska Institutet, the Swedish Research Council, the Ragnar Söderberg Foundation, the Åke Wiberg Foundation, and the LeDucq Foundation. The Figs. 1a, 2j, 4a, 5a, b, 6a, g and Supplementary Fig. 5a were created by F.L.-B. with BioRender.com.

## Author contributions

N.M., I.B.L. and P.-O.B. conceived the idea of the project; N.M., F.L.-B., B.L. developed and verified the methods involved in the project; F.L.-B., N.O.-V., N.M. carried out experiments and analyzed the data; O.B. and M.B. contributed analysis tools; N.M., I.B.L., B.L. supervised and administered the project; P.-O.B., V.M.L., O.B., N.M. acquired the necessary funding; F.L.-B., N.M. wrote the original draft; F.L.-B., N.O.-V., M.B., B.L., O.B., V.M.L., I.B.L., N.M. and P.-O.B. reviewed and edited the manuscript.

## Funding

## Competing interests

P.-O.B. is co-founder and CEO of Biocrine AB, I.B.L. and B.L. are consultants for Biocrine AB. V.M.L. is a co-founder, CEO, and shareholder of HepaPredict AB, as well as a co-founder and shareholder of PersoMedix AB. The remaining authors declare no competing interests.
