## [Peer Review File · Nature Communications]

Intraocular liver spheroids for non-invasive high-resolution in vivo monitoring of liver cell functionREVIEWER COMMENTS

Reviewer #1 (Remarks to the Author):

1. The 3D culture system showed promising features such as increased expression of liver biomarkers and improved microvasculature structures, so it may serve as Intraocular liver spheroids for non-invasive high-resolution in vivo monitoring of liver cell function. However, the manuscript was poorly written, with the results not clearly presented and details of approaches missing. It raises concerns about the soundness of scientific interpretations and the conclusions.

2. Figure 1 lacks quantitative analysis and detailed explanation. The diameters of spheroids should be show.

Also, liver spheroids should be shown with cell viability data.

3. For Figure 1, liver spheroid should be provided for hepatocytes characteristic and microstructure data.

4. Figure 2 said In vivo imaging of labeled red blood cells (red) travelled through intra-spheroid vessels but It is not easy to distinguish vessels in Fig 2. It would be better to mark the blood vessels.

5. In figure 4, I wonder why none of glucose/lipid metabolism pathways were selected as they were so important for liver functions. Also, how the 10 genes were selected for the real-time PCR?

6. I think, Vascularization is achieved mainly with the presence of LSECs. Please add discussion or proof for liver spheroid or host induced vascularization.

7. What is the size of liver spheroid in vitro or invivo? How do you measure spheroid size?

8. What causes blood vessels to grow in the eye?

9. What are the criterion for selecting media in PHS and LSECS mixed culture?

Reviewer #2 (Remarks to the Author):

Dear Authors,

The manuscript is built on the extensive expertise and anterior chamber of the eye (ACE) models already developed by some of the authors. This provides already a good robust platform where to further develop the platform for longitudinal in vivo imaging of liver spheroids at cellular resolution. The conceptual development and need for a cellular resolution transplanted liver spheroids with demonstrated hepatocyte-specific and liver-like physiological functions is sound and clearly presented.

The imaging approach taken to show case the mature hepatocyte response is also confirmed visually, down at cell-cluster level but also by means of molecular assays showing similar cellular- and organ- response to human liver functions at both healthy and disease stage (e.g., hepatosteatosis).

Given the relevance that this platform model could provide in addressing the continuous screening of pharmacological drugs or possible toxins, it is important that all aspects associated with the development and implementation of this will be taken into account to avoid to fall short of information while running the exposure assessment.

Thus, some points need to be improved and presented as the platform model and its primary objectives are at the centre of the 3R ethical discussion. It is obvious that the platform offers a simultaneous multiple spheroids investigation and observation which bring it into the reduction and/or refinement on the need for animal use. This poses several requests for clarification which I would like the authors to take into account in the manuscript revision, since are not presented in the current version:

- Animal welfare, husbandry, sacrifice and behaviour should be presented as part of the in vivo ethical justification for the study. Please also include ethical authorisation and specify if this was a single isolated study or part of a series of and for what intention this was authorised.
- For the animal behaviour the animals weight (e.g., loss or gain) should be presented as this

is part of the animal study records. If blood chemistry was carried out it would be interesting to see the comparison between the study branches and the behaviour.

- For the two points above, the authors could also referenced and draw comparison with their previously developed ACE models, if they wish to include.

- For the in vivo study the injection and imaging setup is briefly described, I believe the readers would benefit from a visual image of the rig and also animal positioning, setting up time and total imaging length time.

- Regarding the upright laser scanning confocal microscope, it is known that ablation is occurring while the sample is imaged. The extent of this is proportional to the laser intensity, acquisition time, pinhole aperture and number of images taken. As the authors are also using different fluorescent dyes with different excitations and emission wavelengths, the technical information should be provided in order to assess if the damaged induced to the host-organ is reversible or irreversible.

- In light of the above point, based on the imaging settings, the upright laser scanning confocal microscope, there is no details on how the spheroids is impacted by every imaging session through to the culture time since these are grown up to 6-months.

- Clarification: please expand on the sentence: "... platform for non-invasive and longitudinal in vivo imaging of mouse liver spheroids in the ACE at cellular resolution...".

- Clarification: please expand on the selection of the cell lineages adopted in this study and future statistical robustness, as the variability among primary cell lines is often very high.

-

With the continuous thrive that 3D biology is offering nowadays, I would also encourage the authors to a broader reflective approach on how the platform will compete against possible alternative platforms which are not as time and labour intensive as the one presented.

The authors have made quite a large number of statements in the final section of the discussion which are in need of some clarification.

Given the authors will address all the above points in details, I would be happy to reconsider the manuscript for acceptance and possible publication.

COMMENTS REVIEWER #1

We thank this reviewer for the consideration of our work and the valuable comments which we addressed in our response.

Comment 1: The 3D culture system showed promising features such as increased expression of liver biomarkers and improved microvasculature structures, so it may serve as Intraocular liver spheroids for non-invasive high-resolution *in vivo* monitoring of liver cell function. However, the manuscript was poorly written, with the results not clearly presented and details of approaches missing. It raises concerns about the soundness of scientific interpretations and the conclusions.

Response: In reference to the writing of the manuscript, we have revised the main text to improve the clarity of the results and conclusions. Specifically, we have substantially edited the Results section related to the transplantation of human liver spheroids, as well as the concluding paragraph of the Discussion. We have moreover added explanations and details to the Methods section. We also included new supplementary information concerning aspects of mouse *in vitro* liver spheroid characterisation, the transplantation of human liver spheroids and *in vivo* imaging settings, which we explain in further detail in the following responses.

Comment 2: Figure 1 lacks quantitative analysis and detailed explanation. The diameters of spheroids should be shown. Also, liver spheroids should be shown with cell viability data.

Response: We have expanded Fig. 1 (by adding panels c and d) to contain the *in vivo* imaging set up and positioning of the mouse, as well as a timeline of *in vivo* imaging preparation and duration. We recognized we have missed the spheroid size information and we have now addressed it in the Supplementary Fig. 1. The average size of the *in vitro* spheroids selected for transplantation is of $248 \pm 13 \mu\text{m}$ (mean \pm SD), calculated by averaging the vertical and horizontal diameters of each spheroid imaged in the 96-well plate in which they are formed plating 1200 cells/well. Our reasons for selecting spheroids of this size are the following: (1) the spheroid size should not be too large to avoid hypoxia and necrotic core, but they should contain enough cells to support cell-cell communications and to allow graft remodelling in the eye, (2) the weight of spheroids of this size allows them to gravitate towards the iris and allow their engraftment, (3) this size is appropriate in relation to transplanting 5-10 spheroids per mouse eye. As to the size of liver spheroids engrafted in the eye post-transplantation, we have calculated the size of the spheroids by *in vivo* imaging, at 3 weeks post-transplantation, when we consider the liver spheroids have reached a stable shape and size. In the *in vivo* situation, we also calculated their size by averaging the vertical and horizontal diameters. We have included these data and the method used, along with a schematic on how we calculate spheroid size both *in vitro* and *in vivo*, in Supplementary Fig. 1e-g.

Regarding the cell viability of *in vitro* liver spheroids prior to transplantation, we agree that this aspect was not addressed. Therefore, we have performed a live/dead cell assay on liver spheroids in culture to determine their cell viability. The spheroids showed high cell viability ($99.2 \pm 0.4 \%$; mean \pm SD) with very few dead cells mainly located to the spheroid surface and no evidence of necrotic core. We have added this new data into Supplementary Fig. 1h. Regarding the cell viability of the liver cells within the grafts *in vivo*, we directly demonstrate the functioning of the hepatocytes through the diverse characterisation experiments in the manuscript (e.g. bile export

and lipoprotein uptake in Fig.3-4), which show that the cells are viable and perform their parenchymal tasks.

Comment 3: For Figure 1, liver spheroid should be provided for hepatocytes characteristic and microstructure data.

Response: Liver spheroids have been widely used over the past decade as a 3D liver *in vitro* tissue model, with many studies that provide in-depth characterisation of this model [1, 2]: In response to this comment, we added a characterisation of our cultured liver spheroids prior to transplantation. Therefore, we have added RNA *in situ* hybridization, showing single-cell spatial gene expression of hepatic markers Albumin and HNF4a (Supplementary Fig. 1a), as well as immunostaining of structural proteins F-actin (phalloidin) and E-cadherin (CDH1) (Supplementary Fig. 1b,c). Moreover, we point out typical bi-nucleated hepatocytes within the spheroid (Supplementary Fig. 1d). After transplantation and engraftment, the liver spheroids have been extensively characterised in the eye throughout the paper, showing the expression of hepatocyte-specific markers (Fig. 3a,b,c), evidence of hepatocyte cell microstructure (Fig. 3d) and proof of hepatocyte functionality by *in vivo* assays (Fig. 4).

Comment 4: Figure 2 said *In vivo* imaging of labeled red blood cells (red) travelled through intra-spheroid vessels but It is not easy to distinguish vessels in Fig. 2. It would be better to mark the blood vessels.

Response: We thank the reviewer for this comment and have made changes to make this experiment clearer. Namely, we have substituted this panel for new representative images of an experiment in which we co-injected the labelled RBCs with lectin, which stains the blood vessels *in vivo* (Fig. 2f). Accordingly, we have also substituted Supplementary Movie 1 with this new experiment.

Comment 5: In figure 4, I wonder why none of glucose/lipid metabolism pathways were selected as they were so important for liver functions. Also, how the 10 genes were selected for the real-time PCR?

Response: As for clarification, we would like to point out that all gene expression data in this study was obtained from bulk RNA sequencing (not real-time qPCR) in spheroids excised from the ACE. The entire dataset will be available in a GEO repository (GSE245944) to search for specific gene set or pathway analysis. In case the reviewers want to access it already they can use this safe link:

<https://eur01.safelinks.protection.outlook.com/?url=https%3A%2F%2Fwww.ncbi.nlm.nih.gov%2Fgeo%2Fquery%2Facc.cgi%3Facc%3DGSE245944&data=05%7C01%7Cnuria.vilarnau%40k.i.se%7Cc228eed427b14a19b86a08dbe9a407d6%7Cbff7eef1cf4b4f32be3da1dda043c05d%7C0%7C0%7C638360662948832091%7CUnknown%7CTWfpbGZsb3d8eyJWljoimc4wLjAwMDAiLCJQljoiv2luMzIiLCJBTiI6Ik1haWwiLCJXVCI6Mn0%3D%7C3000%7C%7C%7C&sdata=EsI%2BOaQNHA8CBkVlHmhc%2BTQfAT9WJl9voEFFy5e6ixo%3D&reserved=0>

and enter the token qnstgwuorxsptgp into the box.

In our Figure 4 (b,f), we selected those 10 genes mined from the bulk RNA sequencing dataset due to their connection to the imaging experiment shown in the same figure (bile acid assay and LDL uptake). We selected those representative genes, which are known to be key players in bile and lipid metabolism from publications focussing on the characterisation of hepatocytes [2-4].

Regarding the lack of gene expression for glucose/lipid metabolism pathways, in Supplementary Fig. 4b we show gene expression data related to glucose and glycogen metabolism, while in Fig. 4f, we show key lipid metabolism genes supported by an *in vivo* assay of LDL-uptake in hepatocytes (Fig. 4g). However, in the main figures, glucose metabolism related genes were not added since our main goal was to show the imaging capabilities of our platform related to different hepatocyte functions. Since we could not find specific probes/assays which can measure this specific aspect *in vivo*, we added a few representative genes derived from the mining of our RNAseq dataset in the Supplementary Fig.4.

Comment 6: I think, Vascularization is achieved mainly with the presence of LSECs. Please add discussion or proof for liver spheroid or host induced vascularization.

Response: We have revised the section of the manuscript on the transplantation of human liver spheroids into recipient immunocompromised mice which, as pointed out by the reviewer, lacked the explanation of the rationale behind the use of primary human liver sinusoidal endothelial cells (LSECs). When transplanting liver spheroids made of purified primary human hepatocytes (PHH), these spheroids failed to engraft and become vascularised in the mouse eye. We attribute this to the purchased PHH being highly purified and therefore lacking supporting cell types, such as endothelial cells or Kupffer cells, both of which are known to secrete pro-angiogenic and growth factors, which promote vascularisation [5, 6]. Additionally, there could be species-specific angiocrine signals which are hindered by the transplantation of human tissue into mouse recipient. Therefore, we hypothesized that the addition of primary human LSECs could facilitate the vascularisation of the graft. This strategy has been used by others, for example, in a study by Takebe et al. [7], iPSC-derived hepatocytes were cultured with supportive mesenchymal and endothelial cells to form 3D-liver buds, which were then transplanted onto the mouse brain, where they successfully engrafted and became vascularised. Thus, we generated liver spheroids from a 2:1 mixture of PHH and LSECs *in vitro* and transplanted them into the ACE, where they became vascularised. Therefore, we agree with Reviewer #1, that the vascularisation of human liver spheroids in the ACE of mice is made possible by the presence of LSECs and we have added this in the Results (Page 5 line 3-12) and Discussion (Page 10 line 13-28).

Comment 7: What is the size of liver spheroid in vitro or invivo? How do you measure spheroid size?

Response: The average size of liver spheroids in culture prior to transplantation is of approx. 250 μm , corresponding to seeding 1200 cells/well and the size is measured by calculating the average of vertical and horizontal diameters. Upon transplantation into the ACE, the spheroid mass can be easily differentiated from the iris tissue due to different backscatter signal intensity, therefore we use the same strategy to measure spheroid size. We have added this new data into the supplementary material, alongside a schematic explanation of how the spheroid size was calculated (Supplementary Fig. 1e-g).

Comment 8: What causes blood vessels to grow in the eye?

Response: Our group has longstanding experience with using the anterior chamber of the eye as a transplantation site for pancreatic islets [8]. In this unique transplantation site, the transplanted tissue becomes vascularised by vessels sprouting from the iris vascular bed. This natural vascularisation of grafts is also seen when transplanting liver cells to other intravital sites, such as into lymph nodes [9] or the brain [10]. We hypothesize that vascularisation in the ACE is induced by the graft's secretion of angiogenic factors, such as VEGF-A. Moreover, immune cells such as macrophages, that we have shown are actively recruited to the graft in the eye, are known to secrete pro-angiogenic factors [11].

Comment 9: What are the criterion for selecting media in PHS and LSECs mixed culture?

Response: For the co-culture of PHH and LSECs, we followed the protocols by Ware et al. and Bale et al, in which primary hepatocytes and LSECs are co-cultured [12, 13]. The composition of the Williams E hepatocyte media we used is optimized to avoid the de-differentiation of primary hepatocytes in culture. In response to this comment, we also provide new immunofluorescence staining of CD31-positive endothelial cells within the spheroid mass (Supplementary Fig. 2d), showing that LSECs survive and are present in the co-culture spheroids prior to transplantation.

COMMENTS REVIEWER #2

General response to Reviewer #2:

We appreciate the positive and constructive criticism from Reviewer #2 and we are confident we provided the additional data and information the referee requested.

Comment 1: Animal welfare, husbandry, sacrifice and behaviour should be presented as part of the *in vivo* ethical justification for the study. Please also include ethical authorisation and specify if this was a single isolated study or part of a series of and for what intention this was authorised.

Response: We agree and modified the first section in the Methods as "Animal welfare, husbandry, behavior and sacrifice" and added an "Ethical authorization" paragraph.

Firstly, we now explain that mice were housed in littermate groups of 2-6 animals and importantly, that there was no need for animals to be in isolation neither prior nor after surgery or *in vivo* imaging sessions. We believe this aspect is crucial for animal welfare, given that mice are social animals. We have also added details on the sacrifice of the animals, which is carried out at the end of the experiment but is not determined by health deterioration of the animal caused by the transplantation surgery or *in vivo* imaging sessions. Additionally, we now reiterate in the Methods section that the operated animals recover quickly from the surgical intervention, with no need of post-operative care, and that the transplantation does not affect their behaviour. We consider normal behaviour to be aspects such as, no increased fighting between males, normal feeding and growth, normal levels of activity and nest-making. Moreover, transplanted animals have been maintained for over 1 year, with no apparent effect to their health or longevity.

Secondly, we cover the ethical authorisation. This study is covered by our group's ethical permit "Studies on the function of hormone-releasing and hormone-stimulated cells and related cells/tissues in normal and diabetic animal models and in transplanted tissues and cells" (n. 6362-2023, previous 16454-2022, 17431-2021, 8822-2020) approved by the Animal Experiment Ethics Committee at Karolinska Institutet. This permission describes the transplantation of microtissues into the ACE, with the objective of establishing platforms for non-invasive *in vivo* imaging of tissues at cellular resolution. The ethical justification for these types of studies is because, beside the research purposes, longitudinal imaging of the same cells in the same subject over time allows the reduction of experimental animals, while at the same time improving the quality of the *in vivo* data, given that we can monitor reaction and progression in individual cells over time. Thus, this study forms part of a larger effort to promote the anterior chamber of the eye as a platform for non-invasive imaging of different tissues and its application in both basic and pre-clinical medical research.

Comment 2: For the animal behaviour the animals weight (e.g., loss or gain) should be presented as this is part of the animal study records. If blood chemistry was carried out it would be interesting to see the comparison between the study branches and the behaviour.

Response: Given that our group has longstanding experience with the implementation of this imaging method, we did not originally record the weight progression of transplanted vs non-transplanted animals. However, we agree that it is important to show the effects of the transplantation surgery and *in vivo* imaging in recipient mice. In addition to recording weight gain/loss, we have monitored non-fasting blood glucose levels in both groups, as an indirect measure of sustained stress and metabolic alterations. We found that neither weight nor blood glucose levels significantly differed between the transplanted and non-transplanted groups over a period of 1-month post-transplantation. Additionally, we performed *in vivo* imaging at 3 weeks post-transplantation in recipient animals, which did not have a negative effect on their health. We have created Supplementary Fig. 6. to include these new data.

Comment 3: For the *in vivo* study the injection and imaging setup is briefly described, I believe the readers would benefit from a visual image of the rig and also animal positioning, setting up time and total imaging length time.

Response: We have added an image showing the *in vivo* imaging set up and positioning of the mouse (Fig. 1c), as well as a timeline (Fig. 1d) depicting the set-up steps and imaging duration.

Comment 4: Regarding the upright laser scanning confocal microscope, it is known that ablation is occurring while the sample is imaged. The extent of this is proportional to the laser intensity, acquisition time, pinhole aperture and number of images taken. As the authors are also using different fluorescent dyes with different excitations and emission wavelengths, the technical information should be provided in order to assess if the damaged induced to the host-organ is reversible or irreversible.

Response: We have created Supplementary Table 2, in which we summarize the settings we use for *in vivo* imaging of the different fluorescent reporter proteins and injectable fluorescent probes.

Comment 5: In light of the above point, based on the imaging settings, the upright laser scanning confocal microscope, there is no details on how the spheroids is impacted by every imaging session through to the culture time since these are grown up to 6-months.

Response: Despite repeated imaging sessions, the laser exposure time per spheroid is limited to 45-60 seconds, in case of capturing a 3D spheroid Z-stack scan (at 600 Hz speed and stacks of 4 μm thickness). If capturing individual images of clusters of cells, a single plane image is taken in less than one second. We believe the impact of laser incidence on the spheroids is negligible, for the following reasons. On the macroscopic level, we see no changes to the appearance of the iris and cornea after the imaging sessions (no signs of redness, irritation or cloudiness of the aqueous humour). In terms of animal welfare and pain, the mice show no signs of pain when awakening from an *in vivo* imaging session (according to the Mouse Grimace Scale; score of 0 after 30 min of recovery). On the microscopic level, the imaging does not induce photobleaching (given we use the same laser power to observe a given fluorophore in repeated imaging sessions), nor do we see any damage to cells or organelle structures, which show intact functionality despite having been imaged multiple times. Moreover, as Reviewer #2 has alluded to, our research group relies on over a decade of experience with this imaging platform to conclude that longitudinal *in vivo* imaging sessions, if performed under our conditions, do not cause photodamage or have a detrimental effect on the transplanted tissue or the recipient animal. To clarify this in the text, we have added a sentence in the Methods section (Page 15 lines 28-29) to explain that photodamage does not occur in the imaged liver spheroids when using our imaging conditions.

Comment 6: Clarification: please expand on the sentence: "... platform for non-invasive and longitudinal *in vivo* imaging of mouse liver spheroids in the ACE at cellular resolution...".

Response: We have rephrased the sentence to more clearly highlight the two main advantages of this imaging platform; the possibility of longitudinal studies and high resolution imaging at single-cell level.

Original sentence: In conclusion, we have established and characterized a platform for non-invasive and longitudinal *in vivo* imaging of mouse liver spheroids in the ACE at cellular resolution.

Revised version: In conclusion, we have established and characterized a platform for *in vivo* imaging of mouse liver spheroids, in which the graft can be imaged non-invasively and repeatedly at different time points in the same animal (longitudinally) at single-cell resolution.

Comment 7: Clarification: please expand on the selection of the cell lineages adopted in this study and future statistical robustness, as the variability among primary cell lines is often very high.

Response: We apologize for the confusion. In this study we did not use any cell lines but only primary cells, isolated directly from adult mouse or human liver. Notably, these primary liver cells are not proliferative. To avoid or reduce inter-donor variability of primary cells (and thus transplanted spheroids) in our experiments, we isolated primary cells from animals of the same strain and selected similar age and sex. In the case of human material, we agree that inter-donor variability can be high. However, as cells can be cryopreserved, it is possible to use material from

the same donor in multiple experiments, which allows to abstract from inter-individual variability.

Comment 8: With the continuous thrive that 3D biology is offering nowadays, I would also encourage the authors to a broader reflective approach on how the platform will compete against possible alternative platforms which are not as time and labour intensive as the one presented.

Response: The ACE is a unique transplantation site in terms of its optical accessibility, and compared to the installation of an abdominal body window [14], the ACE surgery is notably quicker, easier and less invasive for the animal. Therefore, we have not previously commented on this aspect in our Discussion. However, we are not imaging the endogenous liver, and it should not be considered as another form of liver intravital imaging. We do not envision our model as replacement of *in vitro* 3D liver models. Rather, we believe that our platform would be used in combination to high-throughput *in vitro* techniques. For example, as a platform for *in vivo* validation of compounds or therapies, which have previously been explored by *in vitro* studies. We have added this last consideration to the Discussion (Page 11 lines 4-7).

Comment 9: The authors have made quite a large number of statements in the final section of the discussion which are in need of some clarification.

Response: Thanks for the valuable suggestion. We have revised as follows.

Original paragraph: In conclusion, we have established and characterized a platform for non-invasive and longitudinal *in vivo* imaging of mouse liver spheroids in the ACE at cellular resolution. These spheroids preserve liver-like features and wherein hepatocytes retain their overall differentiation and functionality. By different proof-of-concepts experiments we have shown the capabilities of this platform spanning multiple areas of liver research and we foresee a future in both basic research and as a testing-tool for therapeutics in pre-clinical and translational studies.

Revised version: In conclusion, we have established and characterized a platform for *in vivo* imaging of mouse liver spheroids, in which the graft can be imaged non-invasively and repeatedly at different time points (longitudinally) in the same animal at single-cell resolution. The liver spheroids preserve liver-like features and retain hepatic differentiation and functionality. By different proof-of-concepts experiments, we have shown the monitoring capabilities of this platform in different areas of liver research, such as metabolic disease and liver regeneration. We believe this platform will be of great value in both basic research and translational studies.

References

1. Bell, C.C., et al., *Characterization of primary human hepatocyte spheroids as a model system for drug-induced liver injury, liver function and disease*. Sci Rep, 2016. **6**: p. 25187.
2. Oliva-Vilarnau, N., et al., *A 3D Cell Culture Model Identifies Wnt/beta-Catenin Mediated Inhibition of p53 as a Critical Step during Human Hepatocyte Regeneration*. Adv Sci (Weinh), 2020. **7**(15): p. 2000248.
3. Hu, H., et al., *Long-Term Expansion of Functional Mouse and Human Hepatocytes as 3D Organoids*. Cell, 2018. **175**(6): p. 1591-1606.e19.

4. Kim, D.S., et al., *A liver-specific gene expression panel predicts the differentiation status of in vitro hepatocyte models*. *Hepatology*, 2017. **66**(5): p. 1662-1674.
5. Ramirez-Pedraza, M. and M. Fernandez, *Interplay Between Macrophages and Angiogenesis: A Double-Edged Sword in Liver Disease*. *Front Immunol*, 2019. **10**: p. 2882.
6. Poisson, J., et al., *Liver sinusoidal endothelial cells: Physiology and role in liver diseases*. *J Hepatol*, 2017. **66**(1): p. 212-227.
7. Takebe, T., et al., *Vascularized and functional human liver from an iPSC-derived organ bud transplant*. *Nature*, 2013. **499**(7459): p. 481-4.
8. Speier, S., et al., *Noninvasive in vivo imaging of pancreatic islet cell biology*. *Nat Med*, 2008. **14**(5): p. 574-8.
9. Komori, J., et al., *The mouse lymph node as an ectopic transplantation site for multiple tissues*. *Nat Biotechnol*, 2012. **30**(10): p. 976-83.
10. Camp, J.G., et al., *Multilineage communication regulates human liver bud development from pluripotency*. *Nature*, 2017. **546**(7659): p. 533-538.
11. Ribatti, D. and E. Crivellato, *Immune cells and angiogenesis*. *J Cell Mol Med*, 2009. **13**(9A): p. 2822-33.
12. Ware, B.R., et al., *A Cell Culture Platform to Maintain Long-term Phenotype of Primary Human Hepatocytes and Endothelial Cells*. *Cell Mol Gastroenterol Hepatol*, 2018. **5**(3): p. 187-207.
13. Bale, S.S., et al., *Long-term coculture strategies for primary hepatocytes and liver sinusoidal endothelial cells*. *Tissue Eng Part C Methods*, 2015. **21**(4): p. 413-22.
14. Ritsma, L., et al., *Surgical implantation of an abdominal imaging window for intravital microscopy*. *Nat Protoc*, 2013. **8**(3): p. 583-94.

REVIEWERS' COMMENTS

Reviewer #1 (Remarks to the Author):

The manuscript has been well improved, and I think most of my comments have been addressed with either new data or necessary discussions.

Reviewer #2 (Remarks to the Author):

Dear Authors,

Many thanks for addressing all the queries and providing clarification for the many technical aspects of the manuscript in need for the interpretation of the experimental outcomes and possible reproducibility. The new set of improved figures with added experimental rigs, additional controls, histopathology, immunohistochemistry and gene expression are expanding on the depth of the advanced model of the anterior chamber of the eye (ACE) and its potential for the preclinical screening of new therapeutics.

The rationale presented behind the ACE use as a platform for non-invasive imaging of different tissues and its application in both basic and pre-clinical medical research is well presented and supported by the experimental data and evidence of potential relevance produced. The welfare of the animals is now supported by extra data presented in the supplemental information as part of the daily expert handling of the model.

The rationale behind the non-permanent ablative element of the laser exposure during the z-stacks and multiple imaging is presented with additional supporting information.

Nonetheless the authors have missed to produce a critical information which I would encourage to add into the table: the power of the laser and the total energy administered in terms of total energy per single imaging event and also as part of total screening. These are non-trivial information since soft tissue damage to sensitive structure, such as the eye, can happen as part of a single event or as cumulative series of events. I trust these details are at your disposal given the decade of extensive experience and volume of work accumulated.

With regards to conclusions and final statements, I appreciated the effort from the authors in expanding the overall conclusions and promising outlook. I would however encourage the revision of the final conclusive sentence "...We believe this platform will be of great value in both basic research and translational studies..." into the following "...We believe this

platform will be of great value in both basic research and translational studies, within the limitations dictated by latest international directives on the use of animal models.”

I noticed in the manuscript there are no mention to the latest European Directives in animal refinement (given the specific case), which I would encourage to add into the materials and methods as part of the justifications of the animal model selection since these have to be presented at the submission of the ethical license request for the study. As you know, Europe is driving the development of new alternative models to animals thus there is the need for the introduction of a justification behind the use of the ACE model.

Given the promising data presented, I look forward to seeing the outcome of more focused therapeutic investigation towards new NAFLD.

I hope you will see the benefits on making these final changes towards a scientifically robust manuscript.

We thank the reviewers for their comments, which helped to improve the manuscript. We addressed the final questions raised by Reviewer #2 as follows.

- We referred to the latest European Directives in the manuscript in the material and methods under Ethical authorization and justifications (Page 12, line 30) by adding the sentence: The by us established in vivo imaging platform directly addresses two of the 3R principles such as “Reduction” and “Refinement” in compliance with the European Directive 2010/63/EU.
- We changed the last sentence in the Discussion (Page 11, line 26) as suggested by the referee.
- In order to give potential users of our described in vivo imaging platform additional information that allows to reproduce our data using optical systems not identical to ours, we included in the Supplementary Table 2 the lasers power details from the manufacturer as suggested by the reviewer and the power output at the imaging site with our specific settings described in the same table. This was measured using Thorlabs PM100USB Power and Energy Meter Interface with USB Operation combined with Thorlabs S170C Microscope Slide Photodiode Power Meter Sensor (Thorlabs Inc. Newton, New Jersey, U.S.). We avoided introducing the total energy input per screening during our imaging session for several reasons. The energy input is dependent on the number of photons and dwelling time per pixel/voxel and would only be a gross estimation that will not allow to extrapolate information on the possible biological consequences. This number cannot be accurate and is quite variable for every imaging type due to the laser used and the properties of the imaged biological specimen. The fact that our fluorescent probes do not show signs of photobleaching and the biological function monitored is not deteriorating is a good indicator of that the energy input in the system is below a critical level causing persisting damages to the cells/organelles.